# EMERGENT PROPERTIES OF FOVEATED PERCEPTUAL SYSTEMS

## ABSTRACT

We introduce foveated perceptual systems – a hybrid architecture inspired by human vision, to explore the role of a *texture-based* foveation stage on the nature and robustness of subsequently learned visual representation in machines. Specifically, these two-stage perceptual systems first foveate an image, inducing a texture-like encoding of peripheral information – mimicking the effects of *visual crowding* – which is then relayed through a convolutional neural network (CNN) trained to perform scene categorization. We find that these foveated perceptual systems learn a visual representation that is *distinct* from their non-foveated counterpart through experiments that probe: 1) i.i.d and o.o.d generalization; 2) robustness to occlusion; 3) a center image bias; and 4) high spatial frequency sensitivity. In addition, we examined the impact of this foveation transform with respect to two additional models derived with a rate-distortion optimization procedure to compute matched-resource systems: a lower resolution non-foveated system, and a foveated system with adaptive Gaussian blurring. The properties of greater i.i.d generalization, high spatial frequency sensitivity, and robustness to occlusion emerged exclusively in our foveated texture-based models, independent of network architecture and learning dynamics. Altogether, these results demonstrate that foveation – via peripheral texture-based computations – yields a distinct and robust representational format of scene information relative to standard machine vision approaches, and also provides symbiotic computational support that texture-based peripheral encoding has important representational consequences for processing in the human visual system.

## 1 INTRODUCTION

In the human visual system, incoming light is sampled with different resolution across the retinal area, a stark contrast to machines that perceive images at uniform resolution. One account for the nature of this *foveated* (spatially-varying) array in humans is related purely to sensory efficiency (biophysical constraints) (Land & Nilsson, 2012; Eckstein, 2011), e.g., there is only a finite amount of retinal ganglion cells (RGC) that can relay information from the retina to the LGN constrained by the flexibility and thickness of the optic nerve. Thus it is "more efficient" to have a moveable high-acuity fovea, rather than a non-moveable uniform resolution retina when given a limited number of photoreceptors as suggested in Akbas & Eckstein (2017). Machines, however do not have such wiring/resource constraints – and with their already proven success in computer vision (LeCun et al., 2015) – this raises the question if a foveated inductive bias is even necessary for vision at all.

However, it is also possible that foveation plays a functional role at the *representational level*, which can confer perceptual advantages as has been explored *in humans*. This idea has remained elusive in computer vision, but popular in vision science, and has been explored both psychophysically (Loschky et al., 2019) and computationally (Poggio et al., 2014; Cheung et al., 2017; Han et al., 2020). There are several symbiotic examples arguing for the functional advantages of foveation in humans, via functional advantages in machine vision systems. For example, in the work of Pramod et al. (2018), blurring the image in the periphery gave an increase in object recognition performance of computer vision systems by reducing their false positive rate. In Wu et al. (2018)'s GistNet, directly introducing a dual-stream foveal-peripheral pathway in a neural network boosted object detection performance via scene gist and contextual cueing. Relatedly, the most well known example of work that has directly shown the advantage of peripheral vision for scene processing in humans is Wang & Cottrell

| Image | Foveation Transform: $f_*(\circ)$ | Foveated Image |
|---|---|---|
| 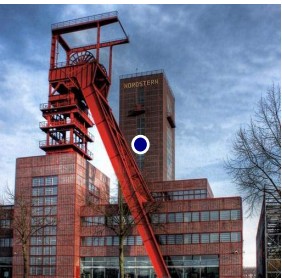 | 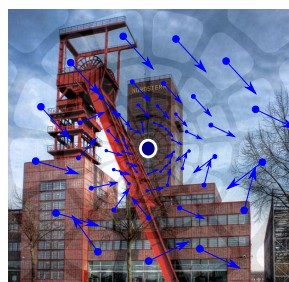 | 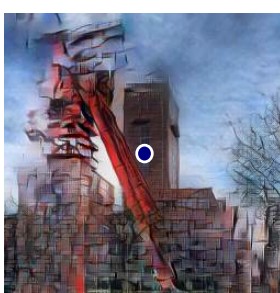 |

Figure 1: A cartoon illustrating how a foveated image is rendered resembling a human visual *metamer* via the foveated feed-forward style transfer model of Deza et al. (2019). Here, each receptive field is locally perturbed with noise in its latent space in the direction of their equivalent texture representation (blue arrows) resulting in *visual crowding* effects in the periphery. These effects are most noticeable far away from the navy dot which is the simulated center of gaze (foveal region) of an observer.

(2017)'s dual stream CNN that modelled the results of Larson & Loschky (2009) with a log-polar transform and adaptive Gaussian blurring (RGC-convergence). Taken together, these studies present support for the *functional* hypothesis of a foveated visual system.

Importantly, none of these studies introducing the notion of texture representation in the periphery – a key property of peripheral computation as posed in Rosenholtz (2016). Testing whether this texture-based coding of the visual periphery is *functional* in any perceptual system is still an open question. Here we address this question directly. Specifically, we introduce *foveated perceptual systems*: these are two-stage (hybrid) systems that have a texture-based foveation stage followed by a deep convolutional neural network. In particular, we will mimic foveation over images using a transform that simulates *visual crowding* (Levi, 2011; Pelli, 2008; Doerig et al., 2019b;a) in the periphery as shown in Figure 1 (Deza et al., 2019), rather than Gaussian blurring (Pramod et al., 2018; Wang & Cottrell, 2017) or compression (Patney et al., 2016; Kaplanyan et al., 2019). These rendered images capture image statistics akin to those preserved in human peripheral vision, and resembling texture computation at the stage of area V2, as argued in Freeman & Simoncelli (2011); Rosenholtz (2016); Wallis et al. (2019).

Thus, our strategy in this paper is to compare these hybrid models' perceptual biases to their non-foveated counterpart through a set of experiments: generalization, robustness to occlusion, image-region bias and spatial frequency sensitivity. A difference from Wang & Cottrell (2017) is that our goal is *not* to implement foveation with adaptive gaussian blurring to fit known results to data (Larson & Loschky, 2009); but rather to explore the emergent representational consequences on scene representation following *texture-based foveation*. While it is certainly possible that in these machine vision systems that only need to categorize scenes, there may be little to no benefit of this texture-based computation; the logic of our approach however, is that any benefits or relevant differences between these systems can shed light into both the importance of texture-based peripheral computation in humans, and could suggest a new inductive bias for advanced machine perception.

## 2 FOVEATED PERCEPTUAL SYSTEMS

We define perceptual systems as *two-stage* with a foveation transform (stage 1, $f(\circ) : \mathbb{R}^D \to \mathbb{R}^D$), that is relayed to a deep convolutional neural network (stage 2, $g(\circ) : \mathbb{R}^D \to \mathbb{R}^d$). Note that the first transform stage is a *fixed* operation over the input image, while the second stage has *learnable* parameters. In general, the perceptual system $S(\circ)$, with retinal image input $I : \mathbb{R}^D$ is defined as:

$$S(I) = g(f(I)) \tag{1}$$

Such two stage models have been growing in popularity, and the reasons these models (including our own) are designed to *not* be fully end-to-end differentiable is mainly to *force* one type of computation into the first-stage of a system such that the second-stage $g(\circ)$ must figure out how to capitalize on such forced transformation and thus assess its $f(\circ)$ representational consequences. For example, Parthasarathy & Simoncelli (2020) successfully imposed V1-like computation in stage 1

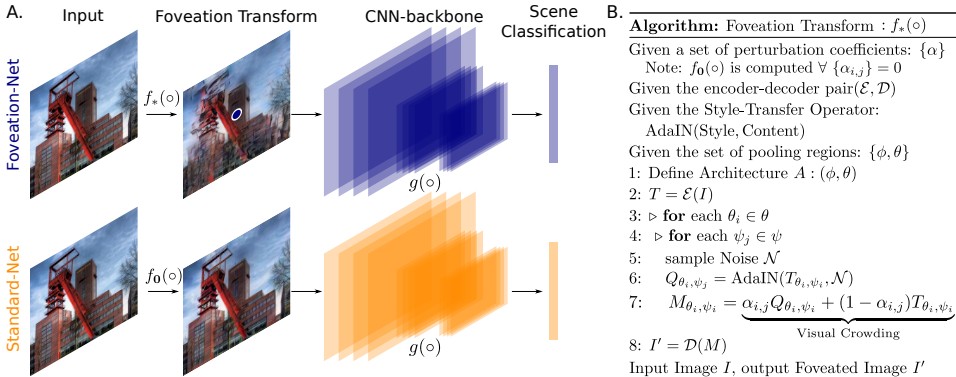

Figure 2: A. Two key Perceptual Systems $S$: Foveation-Net (top row) and Standard-Net (bottom row), where each system receives an image as an input, applies a foveation transform ($f(\circ)$), which is then relayed to a CNN architecture ($g(\circ)$) for scene classification. Foveation-Net uses a visual crowding model with texture computation (shown on right), while Standard-Nets provides a baseline for a perceptual upper-bound for all perceptual systems. B. The algorithm of how a foveated image is generated that enables visual crowding (mainly steps 5-7).

to explore the learned role of texture representation in later stages with a self-supervised objective, and Dapello et al. (2020) found that fixing V1-like computation also at stage 1 aided adversarial robustness. At a higher level, our objective is similar where we would like to force a texture-based peripheral coding mechanism (V2) at the first stage to check if the perceptual system through $g(\circ)$ will learn to pick-up on this newly made representation and make 'good' use of it potentially shedding light on the *functionality* hypothesis for machines and humans. This leads us to further motivate our choice for a single-stream foveated-peripheral pathway (vs dual-stream Wang & Cottrell (2017); Wu et al. (2018)), as we would like to focus on how $g(\circ)$ will differ purely based on the *learning dynamics* driven by SGD rather than architectural constraints (Deza et al., 2020) for the newly transformed input space (foveated : $f_*$ vs non-foveated : $f_0$) to perform a 20-way scene categorization task.

## 2.1 Stage 1: Foveation Transform ($f$)

To model the computations of a foveated visual system, we employed the model of Deza et al. (2019) (henceforth *Foveation Transform*). This model is inspired by the metamer synthesis model of Freeman & Simoncelli (2011), where new images are rendered to have locally matching texture statistics (Portilla & Simoncelli, 2000; Balas et al., 2009) in greater size pooling regions of the visual periphery with structural constraints. Analogously, the Deza et al. (2019) Foveation Transform uses a foveated feed-forward style transfer (Huang & Belongie, 2017) network to latently perturb the image in the direction of its locally matched texture (see Figure 1). Altogether, $f : \mathbb{R}^D \to \mathbb{R}^D$ is a convolutional auto-encoder that is non-foveated when the latent space is un-perturbed: $f_0(I) = \mathcal{D}(\mathcal{E}(I))$, but foveated ($\circ_\Sigma$) when the latent space is perturbed via localized style transfer: $f_*(I) = \mathcal{D}(\mathcal{E}_\Sigma(I))$, for a given encoder-decoder ($\mathcal{E}, \mathcal{D}$) pair.

Note that when calibrating the distortions correctly, the resulting procedure can yield a visual metamer (for a human), which is a carefully perturbed image perceptually indistinguishable from its reference image (Feather et al., 2019). However, importantly in the present work, we exaggerated the strength of these texture-driven distortions (beyond the metameric boundary), as our aim here is to understand the implications of this kind of texturized peripheral input on later stage representations (e.g. following a similar approach as Dapello et al. (2020)). By having an extreme manipulation, we reasoned this would accentuate the consequences of these distortions, making them more detectable in our subsequent experiments.

## 2.2 Stage 2: Convolutional Neural Network backbone ($g$)

The foveated images (stage 1) are passed into a standard neural network architecture. Here we tested two different base architectures: AlexNet (Krizhevsky et al., 2012), and ResNet18 (He et al., 2016). The goal of running these experiments on two different architectures is to let us examine the consequences of foveation that are robust to these different network architectures. Further, this CNN

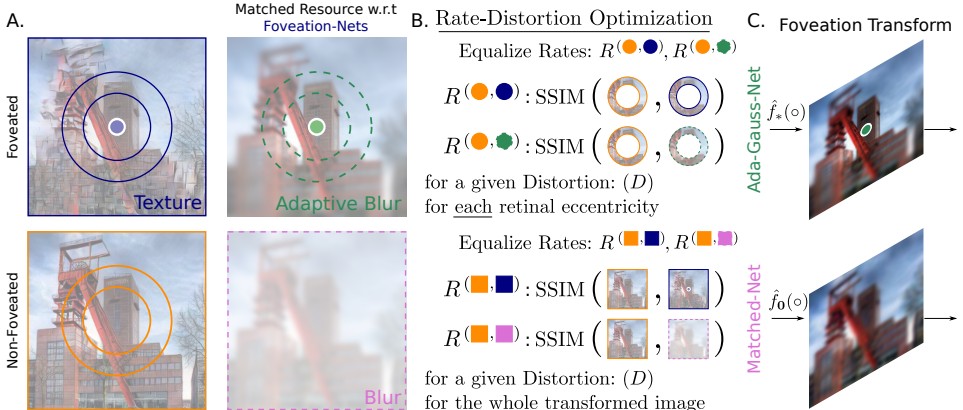

Figure 3: __A.__ Two matched-resource controls to Foveation-Net are introduced. Top, seagreen: adaptive gaussian blurring (Ada-Gauss-Net) emulating RGC convergence-based foveation; Bottom, orchid: uniform blurring emulating a matched-resource non-foveated visual system (Matched-Net). __B.__ A Rate-Distortion Optimization procedure is summarized where we compute the matched-resource images that will define the new foveation transforms: $\{\hat{f}_*, \hat{f}_\mathbf{0}\}$. __C.__ The foveation transforms of both perceptual systems Ada-Gauss-Net (top) and Matched-Net (bottom) do *not* model visual crowding.

backbone ($g : \mathbb{R}^D \to \mathbb{R}^d$) should not be viewed in the traditional way of an end-to-end input/output system where the input is the retinal image ($I$), and the output is a one-hot vector encoding a $d$-class-label in $\mathbb{R}^d$. Rather, the CNN ($g$) acts as a proxy of higher stages of visual processing (as it receives input from $f$), analogous to the 2-stage model of Lindsey et al. (2019).

## 2.3 CRITICAL MANIPULATIONS: FOVEATED VS NON-FOVEATED PERCEPTUAL SYSTEMS

Now, we can define two perceptual systems as the focus of our experiments that must perform 20-way scene categorization: *Foveation-Net*, receives an image input, applies the foveated transform $f_*(\circ)$, and later relays it through the CNN $g(\circ)$; and *Standard-Net*, that performs a non-foveated transform $f_\mathbf{0}(\circ)$, where images are sent through the same convolutional auto-encoder $\mathcal{D}(\mathcal{E}(I))$, but with the parameter that determines the degree of texture style transfer set to 0. Both of these systems are depicted in Figure 2 (__A__).

Anticipating the potential differences between Foveation-Nets and Standard-Nets, a natural next question arises: Will these differences be found with *any* type of spatially-varying processing, or are the effects *specific* to a type of foveation transformation (texture-based coding as linked to area V2 vs adaptive gaussian blurring given retinal constraints). Thus, we created a matched-resource adaptive blurred foveation transform using a Rate-Distortion (RD) optimization framework inspired from Ballé et al. (2016). Figure 3 (B) summarizes our goal and approach: we will have to find a set of standard deviation of the gaussian blurring kernel per retinal eccentricity (the 'distortions' $\mathcal{D}$), such that we can render a perceptually matched adaptive gaussian blurred image – in reference to the compressed image from Standard-Nets – that matches the perceptual transmission 'rate' $\mathcal{R}$ of Foveation-Nets via the SSIM perceptual metric (Wang et al., 2004) – also in reference to Standard-Nets.

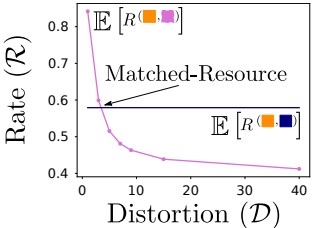

Figure 4: RD-Optimization.

Perceptual Systems trained on these images (which define a new function $\hat{f}_*$), provide a matched-resource control to Foveation-Net that we dub *Ada-Gauss-Net*. This same optimization procedure can be extended to the entire image to also develop a matched-resource control for a non-foveated visual system (*Matched-Net*, $\hat{f}_\mathbf{0}$: Figure 4), as one could argue that Standard-Nets may be at an unfair advantage to Foveation-Nets since their foveal regions are identical and the retinal sensors have not been properly re-distributed (Cheung et al., 2017) & Figure 6 (A).

Altogether these 4 perceptual systems help us answer three key questions as shown in Figure 5 (A.): 1) Foveation-Net vs Standard-Nets will tell us how a texture-based foveation mechanism will compare to its undistorted perceptual upper-bound. 2) Foveation-Nets vs Ada-Gauss-Nets will tell

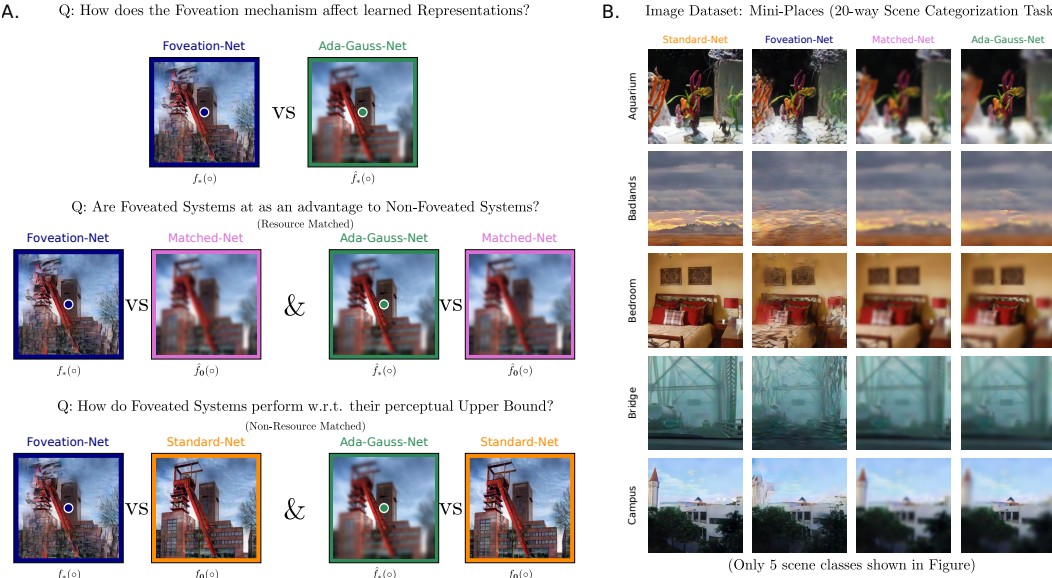

Figure 5: A. Three critical questions that our 4 conditions will help us answer pertaining the role of foveated (spatially-varying) perceptual systems and their comparisons to their non-foveated counterparts. B. Five example images from the 20 scene categories are shown, after being passed through the first stage of each perceptual system.

us if any potentially interesting pattern of results is due to the *type/stage* of foveation. This will help us measure the contributions of the crowding mechanism and texture computation vs adaptive gaussian blurring; 3) Foveation-Net vs Matched-Net will tell us how do these perceptual systems (one foveated, and the other one not) behave when allocated with a fixed number of perceptual resources. In addition our supplementary material includes an extended analysis that includes 2 more highly relevant systems such as Foveation-Aug-Net (a foveated system with the crowding operator coupled with eye-movements – a biological proxy of data-augmentation) and Standard-Aug-Net (a non-foveated system trained with popular data-augmentation mechanisms).

## 3 METHODS

All models were trained to perform 20-way scene categorization. The scene categories were selected from the Places2 dataset (Zhou et al., 2017), with 4500 images per category in the training set, 250 per category for validation, and 250 per category for testing. The categories included were: aquarium, badlands, bedroom, bridge, campus, corridor, forest path, highway, hospital, industrial area, japanese garden, kitchen, mansion, mountain, ocean, office, restaurant, skyscraper, train interior, waterfall. Samples of these scenes coupled with their foveation transforms can be seen in Figure 5 (B.).

**Training:** Convolutional neural networks of the stage 2 of each perceptual system were trained which resulted in 40 networks *per architecture*: 10 Foveation-Nets, 10 Standard-Nets, 10 Matched-Net, 10 Ada-Gauss-Net; totalling 80 trained networks to compute relevant error bars shown in all figures (standard deviations, not standard errors) and to reduce effects of randomness driven by the particular network initialization. All systems were paired such that their stage 2 architectures $g(\circ)$ started with the same random weight initialization prior to training. Other parameters in training via backpropagation were: learning rates: $\eta = 0.001$ AlexNet, $\eta = 0.0005$ ResNet18 – with no learning rate decay for both architectures, and batch size: 128. We trained the AlexNet architectures for each perceptual system $(g(\circ))$ up to 360 epochs and each ResNet18 architecture up to 180 epochs. In the main body of the paper we will show our results for AlexNet at 270 epochs. All effects reported are reproduced across both architectures (AlexNet, ResNet18) and multiple epochs (180,270,360); (90,120,180) respectively in the Supplementary Material which suggests that these effects are independent of learning dynamics.

**Testing:** The networks of each perceptual system were tested on the same type of image distribution they were trained on.

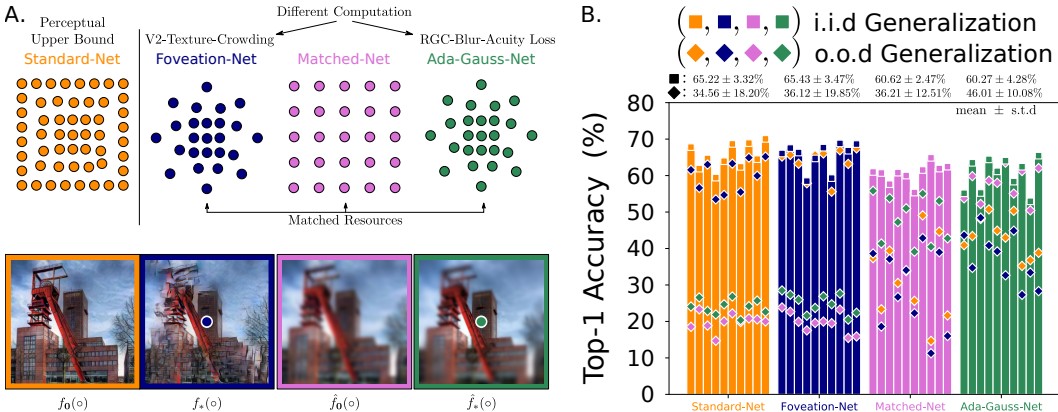

Figure 6: A. A legend illustrating the 4 key perceptual systems in all our experiments with their differences and similarities: Standard-Net, Foveation-Net, Matched-Net and Ada-Gauss-Net. B. Scene Categorization Accuracy: Foveation-Net has the highest i.i.d generalization while Ada-Gauss-Net has the greatest o.o.d generalization – both perceptual systems are *foveated* (spatially-varying).

## 4 EXPERIMENTS

In the following section we will discuss results found with regards to i.i.d. & o.o.d. generalization, robustness to occlusion, central image bias and spatial frequency sensitivity. All results shown in this section use AlexNet as $g(\circ)$ for the second stage of each perceptual system. The same pattern of results are shown with a ResNet18 as stage 2 – these extended results coupled with learning dynamics, as well as extra calculations and additional controls, can be accessed in the Supplementary Material.

### 4.1 GENERALIZATION

We first examined the generalization capacity for our 4 network types. Given that Foveation-Nets receive distorted inputs in the periphery one could expect them to do worse compared to Standard-Nets (which operate over full resolution, untexturized images) and on par with Matched-Nets (which have with perceptually matched resources as Foveation-Nets). On the other hand, it is possible that the texture-based image encoding could confer a functional advantage, in which case Foveation-Nets would do better. If this is indeed the case, we would also see an advantage of Foveation-Nets over Ada-Gauss-Net; but if their performance is the same, we can conclude that *any* type of spatially-varying processing operator can enhance scene categorization performance. Additionally, a somewhat contrived but interesting experiment, we will also conduct is the out-of-distribution (o.o.d) case in which each perceptual system is shown stimuli type that they never encountered during training – in particular stimuli that is the input following the first stage of the *other* perceptual systems.

Result 1: Figure 6(B.) shows that a foveated system with texture-based computation in the periphery (Foveation-Net) *has greater i.i.d generalization* than its matched resource non-foveated system (Matched-Net) and a foveated system with adaptive gaussian blurring (Ada-Gauss-Net); and also performs on par with an un-matched resource non-foveated system (Standard-Net). For an o.o.d generalization task where systems are shown o.o.d images from another system, we find that *drops in performance for foveated perceptual systems are lower compared to their non-foveated counter-part* (Foveation-Net (↑) vs Standard-Net (↓); Ada-Gauss-Net (↑) vs Matched-Net (↓)); and that overall Ada-Gauss-Nets *has greater o.o.d generalization compared to all other systems*, while Foveation-Net struggles to cross-generalize to low pass spatially transformed images compared to Ada-Gauss-Net – a result that ties to the experiments of Section 4.4.

### 4.2 ROBUSTNESS TO OCCLUSION

We next examined how all perceptual systems could classify scene information under conditions of visual field loss, either from the center part of the image (scotoma), or the periphery (glaucoma). This manipulation lets us examine the degree to which the representations are relying on central *vs* peripheral information to classify scene categories. For the foveal-occlusion conditions, we superimposed a central gray square on each image from the testing image set, with 8 levels of

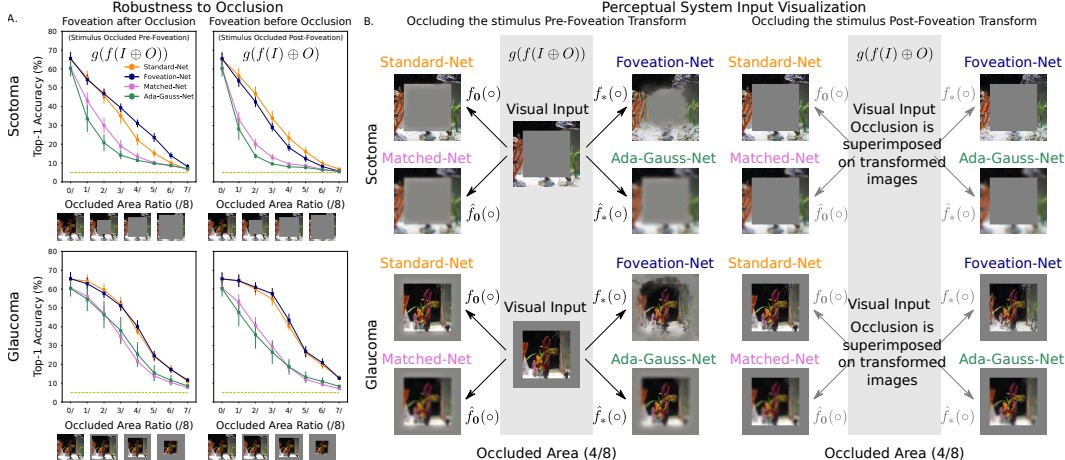

Figure 7: A. The Robustness curves for image occlusion ($\circ \oplus O$) when the Foveation Transform is On (as the stimuli is shown before the transform), and when the Foveation Transform is Off (stimuli shown post-transform). Foveation-Nets performs has greater robustness to occlusion across all matched-resource systems, and also bumps it's performance due to *visual crowding*. B. A visualization of how the visual input is transformed for each perceptual system given their respective Foveation transform ($f$). Left: The effects of crowding are noticeable for Foveation-Nets (but not Ada-Gauss-Nets) as it increases visual area. Right: Even without the aid of visual crowding, the learned texture representation yields greater robustness with matched occluded areas for Foveation-Nets.

increasing size. For the peripheral-occlusion condition, we superimposed a gray box over the outter image boundaries, with area-matched levels of occlusion. These images were then passed through all of the trained models to compute scene categorization performance. Accuracy was measured at each level of occlusion and area under the curve was computed as an index of robustness to occlusion.

Figure 7 (top) shows a summary of these results. Overall Foveation-Nets showed more robustness to central/scotoma occlusion to all other systems: un-matched non-foveated (Standard-Nets), matched-resource foveated system with adaptive gaussian blurring (Ada-Gauss-Net), and matched-resource non-foveated systems (Matched-Net). This result is similar to peripheral/glaucoma occlusion with the exception of Foveation-Nets being on par with non-resource matched Standard-Net. To further explore these results, we visualized what the image inputs look like after the foveation stage, across all systems. Figure 7 (B. left) reveals critical information:

Result 2a: The effect of the stage 1 crowding operation ($f_*(\circ)$) of Foveation-Net *exclusively reduces* the occluded area in the periphery – a stark contrast compared to all other systems. By filling in the image with texture statistics due to the foveation stage ($f_*$) – Foveation-Net (but not Ada-Gauss-Net) are able to boost classification performance by using greater visual information and/or the peripheral texturized representations that Standard-Nets & Matched-Nets do not have access to.

But is this the main factor accounting for the higher robustness of the foveated systems? In other words: is it driven by greater computable visual area that has been filled in from the foveation stage; Or is the benefit driven by the learned texturized representations in the subsequent deep convolutional neural network? To examine this question, we occluded either the center or periphery of the images, but *after* the stage 1 computation (see Figure 7 (A., right)). In this way, the area of occlusion was matched for the second stage classification task.

Result 2b: Across the matched-resource systems and matched-occluded area condition: Foveation-Net has greater robustness to occlusion than Matched-Net and Ada-Gauss-Net. This puzzling result for both scotomal and glaucomal occlusion, not only suggests that the learned representations in Foveation-Net is *different* than Ada-Gauss-Net, but that Foveation-Net has learned a locally distributed representation vs a *globally distributed* representation (Ada-Gauss-Net). In addition as Foveation-Nets and Ada-Gauss-Nets only differ in peripheral computation, this suggests that a texture-based encoding mechanism provides a representational advantage than an adaptive blurred representation when information is unavailable, occluded or removed.

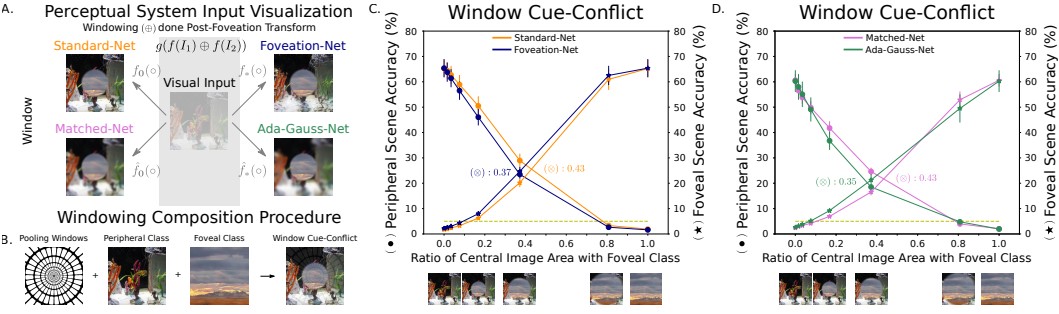

Figure 8: A. A visualization of the stimuli type that is shown to each perceptual system consists of a smoothed composition of a foveal image (*e.g.* badlands) mixed with a peripheral image (*e.g.* aquarium). B. The composition procedure relies on using the same log-polar pooling windows over which the foveated images were rendered. C.,D. Foveated Perceptual Systems – independent the computation type (Ada-Gauss-Net, Foveation-Net) – show stronger biases to classify scenes with the foveal region; a result also observed in humans (Larson & Loschky, 2009; Loschky et al., 2019).

## 4.3 WINDOW CUE-CONFLICT EXPERIMENT

It is possible that foveated systems weight visual information strongly in the foveal region than the peripheral region as hinted by our occlusion results (the different rate of decay for the accuracy curves in the Scotoma and Glaucoma conditions) – and due to the fact that the information in the periphery is texturized/blurred? To assess such difference, we conducted an experiment where we created a windowed cue-conflict stimuli where we re-rendered our set of testing images with one image category in the fovea, and another one in the periphery. All cue-conflicting images were paired with a different class (*ex:* aquarium with badlands). We then systematically varied the fovea-periphery visual area ratio & examined classification accuracy for both the foveal and peripheral scenes (Figure 8).

Result 3: We found that Foveation-Nets and Ada-Gauss-Nets maintained higher accuracy for the foveal scene class than do Standard-Nets and Matched-Nets, as the amount of competing peripheral information increased – and vice versa for the peripheral scene class. A qualitative way of seeing this foveal-bias is by checking the foveal/peripheral ratio where these two accuracy lines cross. The more leftward the cross-over point ($\otimes$), the higher the foveal bias. Thus, Foveation-Nets have learned to weigh information in the center of the image more when categorizing scenes – a similar finding to Wang & Cottrell (2017) indeed as Ada-Gauss-Net vs Matched-Net shows the exact same pattern of results. Thus, these results indicate that the *spatially varying computation from center to periphery* (evident in both Foveation-Nets and Ada-Gauss-Nets) is mainly responsible for the development of a center image bias. It is possible that one of the functions of any spatially-varying coding mechanisms in the visual field is to *enforce* the perceptual system to attend on the foveal region – avoiding the shortcut of learning to attend the entire visual field (Geirhos et al., 2020).

## 4.4 SPATIAL FREQUENCY SENSITIVITY

We next examined differences of learned feature representations that are more reliant on low or high spatial frequency information, at the second stage of visual processing (post-foveation) across all systems. To do so, we filtered the testing image set at multiple levels to create both high pass and low pass frequency stimuli and assessed scene-classification performance over these images for all models, as shown in Figure 9 (A.). Low pass frequency stimuli ($\mathcal{F}_L$) were rendered by convolving a Gaussian filter of standard deviation $\sigma = [0, 1, 3, 5, 7, 10, 15, 40]$ pixels on the foveation transform ($f_\mathbf{0}, \hat{f}_\mathbf{0}, f_*, \hat{f}_*$) outputs. Similarly, the high pass stimuli ($\mathcal{F}_H$) was computed by subtracting the reference image from its low pass filtered version with $\sigma = [\infty, 3, 1.5, 1, 0.7, 0.55, 0.45, 0.4]$ pixels and adding a residual. These are the same values used in the experiments of Geirhos et al. (2019).

Result 4: We found that Foveation-Nets and Standard-Nets were more sensitive to High Pass Frequency information, while Ada-Gauss-Nets and Matched-Nets were sensitive to Low Pass Frequency stimuli. This suggests that foveation via adaptive gaussian blurring (Ada-Gauss-Nets) may implicitly contribute to scale-invariance as also shown in Poggio et al. (2014); Cheung et al. (2017); Han et al. (2020). Additionally, an intriguing possibility is that Foveation-Nets' peripheral texture representation may potentially support a biologically-plausible mechanism of a shape bias as argued in Figure 9.

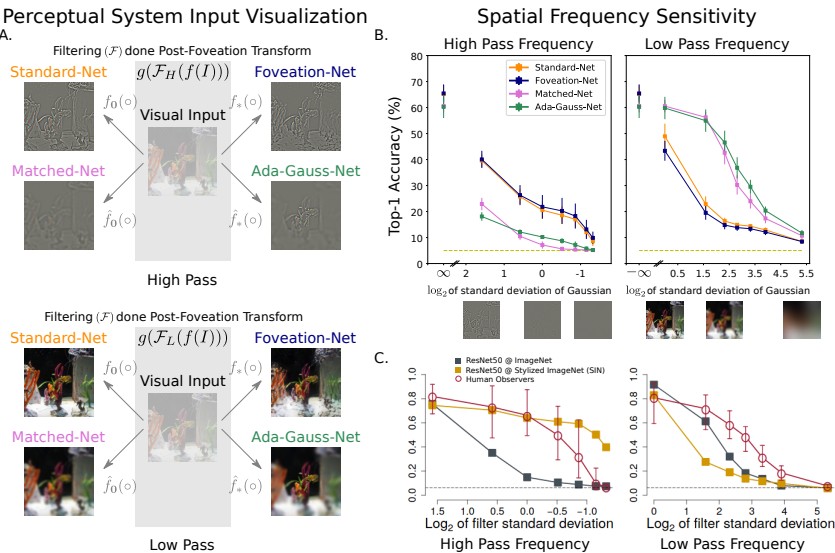

Figure 9: A. Sample images from the foveated and non-foveated images and how they change as a function of spatial frequency filtering at the post-foveation stage. B. Foveation-Nets have nominally greater sensitivity to high spatial frequency filtered stimuli than Standard-Nets, and both of these systems present notably higher sensitivity to high spatial frequencies than Matched-Nets and Ada-Gauss-Nets. Conversely, this pattern is reversed for low pass frequency stimuli. C. This suggests Foveation-Nets's crowding-like computation may naturally enforce a *shape-bias* given it's high pass spatially frequency tolerance as these Spatial Frequency curves show similar trends as Geirhos et al. (2019)'s Stylized ImageNet (SIN) that were trained on objects (adapted and redrawn).

## 5 DISCUSSION

The present work was motivated by the broad question of the functional role of a texture-based peripheral representation. We specifically examined whether a texture-based peripheral encoding mechanism yielded any perceptual advantages and distinctive representational signatures in second-stage deep neural networks. We found that when comparing Foveation-Nets to their matched-resource models that differed in computation: Ada-Gauss-Nets (foveated w/ adaptive gaussian blur) and Matched-Nets (non-foveated w/ uniform blur) – that peripheral texture encoding did lead to specific computational signatures, particularly in robustness to occlusion. We also found that foveation (in general) seems to induce a *focusing mechanism*, servicing the foveal/central region, while the texture-based computation still preserves *high spatial-frequency selectivity* – this last result is likely due to weight-sharing that does not give room to low-spatial frequency tuned filters to naturally emerge as in Wang & Cottrell (2017).

The particular consequences of our foveation stage raises interesting future directions about what computational advantages could arise when trained on object categorization, as objects are typically centered in view and have different hierarchical/compositional priors than scenes (Zhou et al. (2014); Deza et al. (2020)) in addition to different processing mechanisms (Renninger & Malik (2004)). Specifically, one intriguing possibility is that our *foveated* representational signatures may induce more shape sensitivity for object recognition (rather than texture sensitivity; Geirhos et al. (2019); Hermann et al. (2020)), and may amplify the perceptual differences we identified across foveated and non-foveated systems.

Finally, a future direction is investigating the effects of crowding to adversarial robustness. Motivated by the recent work of Dapello et al. (2020) which has shown promise of adversarial robustness via enforcing stochasticity and V1-like computation by matching the Nyquist sampling frequency of these filters (Serre et al., 2007) in addition to a natural gamut of orientations and frequencies as studied in De Valois et al. (1982), it raises the question of how much we can further push for robustness in hybrid perceptual systems like these, drawing on even *more* biological mechanisms. Both Luo et al. (2015) and Reddy et al. (2020) have already taken steps in this direction by coupling fixations with a spatially-varying retina. However, the computational impact of visual crowding and texture-based computation on adversarial robustness is an open question.

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

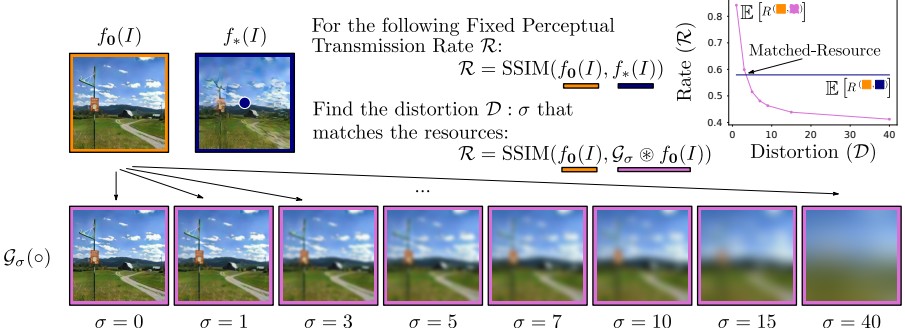

Figure 10: A full explanatory diagram of the Rate-Distortion Optimization Procedure inspired from both Ballé et al. (2016) and Deza et al. (2019). The goal is to find the equivalent 'perceptual transmission rate' for a given distortion $\sigma$ to find a matched-resource perceptual input for Foveation-Nets that is non-foveated (Figure 4). This optimization produces Matched-Nets, a perceptual system that receives as input uniformly blurred images as a way to quantify the expense of computing lower frequencies given retinal ganglion cell re-distribution in as if it were to occur in humans.

# 6    SUPPLEMENTARY MATERIAL

## 6.1    DESCRIPTION OF FOVEATED PERCEPTUAL SYSTEMS

**Foveation-Net**: We adjusted the parameters of the foveation transform to have stronger distortions in the periphery that can consequently amplify the differences between a foveated and non-foveated system. This was done setting the rate of growth of the receptive field size (scaling factor) $s = 0.4$. Thus, when foveation is *on* at the previous scaling factor, we will abbreviate $f$ as $f_*$.

This value was used instead of $s = 0.5$, given that experiments of Freeman & Simoncelli (2011); Deza et al. (2019) has shown that this scaling factor yields a match with physiology but only when human observers are psychophysically tested *between* pairs of synthesized/rendered image metamers. Works of Wallis et al. (2017; 2019); Deza et al. (2019) have suggested that the when comparing a non-foveated *reference image* to it's foveated crowded version, the scaling factor is actually much smaller than 0.5 (0.24, or in some cases as small as 0.20; See Table 1). We thus selected a smaller factor of $s = 0.4$ (that is still metameric to a human observer between synthesized pairs), as smaller scaling factors significantly reduced the crowding effects. Ultimately, the selection of this value is not critical in our studies as we are not making any comparative measurements to human psychophysical experiments (See Deza & Eckstein (2016) for an example where matching scaling factors is critical).

**Standard-Net**: We use the same foveation transform at the foveation stage for Standard-Net but set the scaling factor set to $s = 0$. In this way, any potential effects of the compression/expansion operations of the foveation stage in the image are matched between the Foveation-Nets and Standard-Nets. Thus, the only difference after stage 1 is whether the image statistics were texturized in increasingly large pooling windows (Foveation-Nets), or not (Standard-Nets). Extending our notation, no foveation in stage one will be abbreviated as $f_0$.

Note however, that Standard-Nets are not matched-resource non-foveated controls – Standard-Nets only provide the control for comparing purely the effects of crowding (Foveation-Net) while potentially using *more* resources as both the foveal regions of both transforms remain intact, and a matched resource control should redistribute the retinal ganglion cell convergence uniformly as explored in Cheung et al. (2017). In fact, the matched-resource control that is also non-foveated is Matched-Net as described earlier in the paper, and more in detail as follows.

**Matched-Net**: Matched-Net provides a non-foveated resource matched control with respect to Foveation-Net. This perceptual system is essentially computed via finding the optimal standard deviation $\sigma$ of the filtering kernel $\mathcal{G}_\sigma$ as shown in Figure 10. This distortion image is computed via the convolution ($\circledast$) of the image $f_0(I)$ with the low-pass filter: $\mathcal{G}_\sigma$. Here, Wang et al. (2004)'s SSIM is our idea candidate as it will take into consideration the luminance, contrast and structural changes locally for the entire image and pool them together for an aggregate perceptual score (and also the rate $\mathcal{R}$) that is upper bounded by 1 and correlated with human perceptual judgments. As

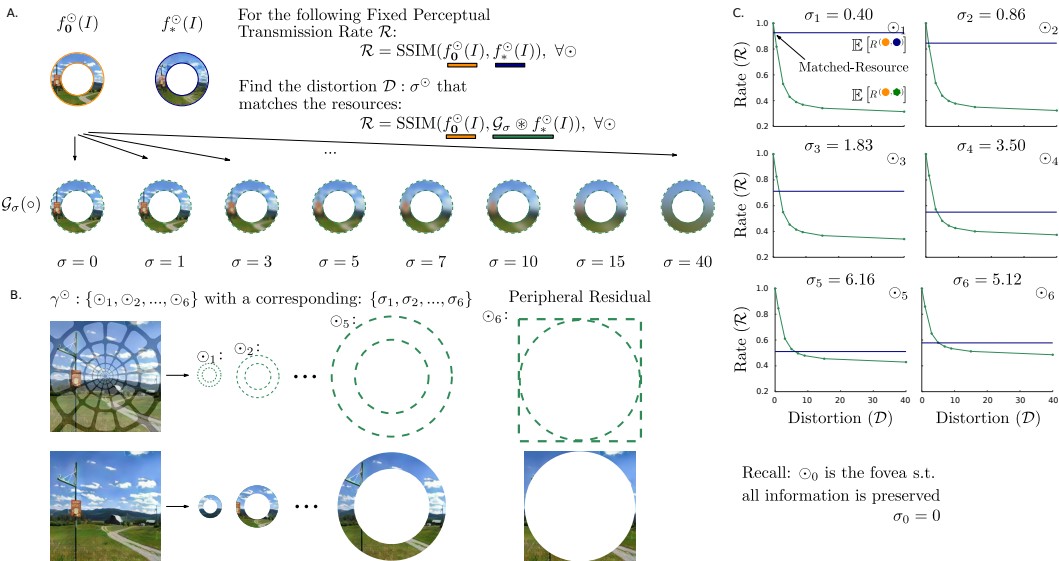

Figure 11: A. The full explanatory diagram (continued) of the Rate-Distortion Optimization Procedure from Figure 10 adapted for Ada-Gauss-Nets. B. The goal is to find the equivalent 'perceptual transmission rate' for a given distortion $\sigma$ to find a matched-resource perceptual input for Foveation-Nets that is foveated but with adaptive gaussian blurring, *i.e.* we must find the standard deviation of the gaussian blurring kernel which is computed over a set of eccentricity rings that have been windowed with cosine functions. C. The full Rate-Distortion curves as a function of retinal eccentricity.

SSIM operates on the luminance of the image, all validation images over which the RD curve (right) was computed were transformed to grayscale to find the optimal standard deviation ($\sigma = 3.4737$).

**Ada-Gauss-Net**: Is a foveated perceptual system that receives Rate-Distortion optimized images that have been blurred with different standard deviations of the gaussian kernel $\mathcal{G}_\sigma$ as a function of retinal eccentricity. We picked the same eccentricity rings (or pooling regions) as Foveation-Nets given that we did not want to include a potential effect that is driven by differences in receptive fields rather than differences in type of computation. Figure 11 shows the full set of distortion strengths ($\sigma$) of each receptive field ring to match the transmission rate of the Foveation Transform ($f_*(\circ)$) that computes crowding.

There are other alternatives to potentially find the set of standard deviation coefficients that are not driven by a rate-distortion optimization procedure. One possibility could have been to find a mapping between pixels and degrees of visual angle as done in Pramod et al. (2018) and derive the coefficients by fitting a contrast sensitivity function given the visual field. While this approach is appealing, the coefficients for object recognition (ImageNet Russakovsky et al. (2015)) can not be extended to scenes (Places Zhou et al. (2017)). In addition, the coupling of the RD-optimization with SSIM provides a perceptual guarantee to compare Ada-Gauss-Net with Foveation-Net or Matched-Net.

**Foveation-Aug-Net** (Supplementary Control): Eye-movements can be seen as a biologically motivated type of data-augmentation strategy. More generally, the goal of this additional control is to test if eye-movements coupled with a texture-based foveated operator will vary the strength of our results in any possible dimension: generalization, robustness, center bias, spatial frequency sensitivity. To test this, we created an enhanced version of Foveation-Net where the deep neural network at stage 2 receives a foveated image of a variable fixation point from one of 9 points from a fixed $3 \times 3$ grid. A schematic of the type of inputs Foveation-Aug-Nets receives can be seen in Figure 12.

Statistical testing (paired t-test, n.s.) revealed no strong differences between Foveation-Nets and Foveation-Aug-Nets, likely due to the fact that scene recognition can be done independent of fixation point (Henderson & Hollingworth, 1999; Oliva & Torralba, 2001). However there are nominal positives differences for Foveation-Aug-Nets over Foveation-Nets, and it is possible that the contribution of eye-movements in learning a new representation is redundant in scene recognition, and that re-computing these experiments for an *object classification task* may yield stronger perceptual biases.

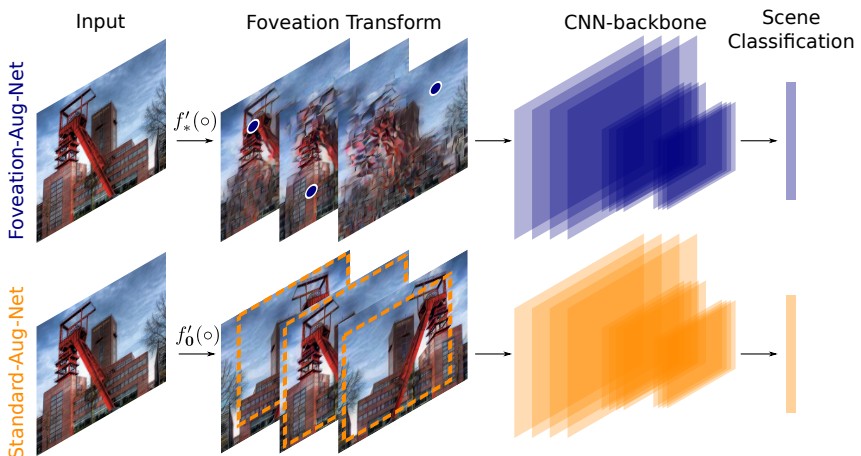

Figure 12: Two additional Perceptual Systems, that serve as data-augmentation controls: Foveation-Aug-Net (top row) and Standard-Aug-Net (bottom row), where each system receives an image as an input, applies a foveation transform ($f'(\circ)$) coupled with a data-augmentation procedure such as eye-movements (Foveation-Aug-Nets) or random cropping + resizing + horizontal mirroring (Standard-Aug-Nets). Once again, these newly transformed image representations are then relayed to a CNN architecture ($g(\circ)$) for scene classification.

**Standard-Aug-Net** (Supplementary Control): In analogy to the previous augmentation condition, Standard-Aug-Net enhances our Standard-Net model with artificial data-augmentation regimes such as random cropping ($0.7 - 1.0$ of area), resizing, and a $0.5$ chance of horizontally flipping the image at training. Standard-Aug-Nets was motivated by 2 main questions: 1) How will actual computer vision systems trained with common data-augmentations behave in reference to other perceptual systems?; 2) Is an augmented non-foveated system asymptotically equal to a foveated system? A schematic of the type of inputs Standard-Aug-Nets receives can be seen in Figure 12.

Several experiments as we will show through-out the Supplementary section, have shown remarkable perceptual similarities between Standard-Aug-Nets and Matched-Nets. This should come at no surprise given that Standard-Aug-Nets received randomly cropped and up-sampled images (thus simulating a type of blur through loss of resolution in the up-sampling stage). What is perhaps most interesting however is that random cropping, up-sampling and horizontal mirroring *did not* contribute to a stronger center bias or to a high spatial frequency sensitivity which is correlated to a shape-bias (See Figure 24 from the extended results of Spatial Frequency sensitivity). These are findings that differ from the results of Hermann & Kornblith (2019); Hermann et al. (2020) that suggest that data-augmentation enhances a shape bias for object recognition. However, recall that scene recognition and object recognition processing mechanisms differ (Renninger & Malik, 2004; Zhou et al., 2014; Deza et al., 2020) – so our findings may in fact be complimentary. This is encouraging as it supports our initial conjectures of the impact of foveation on scenes vs objects, and that stronger pattern of results may emerge when training on such different image input distributions, as mentioned in the Discussion (Section 5).

### 6.2 METHODS (EXTENDED)

All models were trained to perform 20-way scene categorization. The scene categories were selected from the Places2 dataset (Zhou et al., 2017), with 4500 images per category in the training set, 250 per category for validation, and 250 per category for testing. The categories included were: aquarium, badlands, bedroom, bridge, campus, corridor, forest path, highway, hospital, industrial area, japanese garden, kitchen, mansion, mountain, ocean, office, restaurant, skyscraper, train interior, waterfall.

Samples of these scenes coupled with their foveation transforms for each perceptual system can be seen in Section 6.4.

**Training:** Convolutional neural networks of the stage 2 of each perceptual system were trained which resulted in 60 networks *per architecture*: 10 Foveation-Nets, 10 Standard-Nets, 10 Matched-Net, 10 Ada-Gauss-Net, 10 Foveation-Aug-Nets, 10 Standard-Aug-Nets; totalling 120 trained networks to compute relevant error bars shown in all figures (standard deviations, not standard errors) and to reduce effects of randomness driven by the particular network initialization. All systems were paired such that their stage 2 architectures $g(\circ)$ started with the same random weight initialization prior to training, thus the randomness was purely imposed by SGD. This procedure also facilitated our statistical testing via paired t-tests as the model systems began their visual diet with the same set of random weights, thus reasonably 'pairing' them up for statistical analysis when necessary. Other parameters in training via backpropagation were: (learning rates: $\eta = 0.001$ AlexNet, $\eta = 0.0005$ ResNet18 – with no learning rate decay for both architectures), and batch size: 128. We trained the AlexNet architectures for each perceptual system ($g(\circ)$) up to 360 epochs and each ResNet18 architecture up to 180 epochs to study the learning dynamics that can be seen throughout the entire Supplementary Material showing that the main effects and patterns of results reported in the paper are preserved throughout different epochs in training.

**Validation**: Validation images were used exclusively to: 1) visualize the convergence of the loss function for each perceptual system (See Figure 13); 2) compute quantities in our experiments that if otherwise may introduce biases. Examples of these quantities are: the rate-distortion optimization curves for Matched-Nets and Ada-Gauss-Nets; the average channel intensity for each image across all perceptual systems as a sub-step to compute the residual in high-pass spatial frequency filtering (See Section 6.9).

**Testing:** Foveation-Nets were only evaluated on center fixation images at inference time. Standard-Nets were tested on non-foveated images (the output of the foveation transform with scaling factor set to zero). Matched-Nets were evaluated on uniformly blurred images that were gaussian filtered with the same standard deviation coefficient that they were trained on. Ada-Gauss-Nets were tested on adaptive gaussian blurred images that held the same collection of standard deviation coefficients over retinal eccentricities that they were trained on. Standard-Aug-Nets were tested on the same images as Standard-Nets. Foveation-Aug-Nets were tested only on center fixation images (the same images as Foveation-Nets were tested on).

No data-augmentation or multi-crop averaging was performed at inference for any of the networks. No data-augmentation was performed at training for each network either (except for Foveation-Aug-Nets + Standard-Aug-Nets).

All networks performed mean and standard contrast normalization both at training and testing with values set to mean: $(0.485, 0.456, 0.406)$, std: $(0.229, 0.224, 0.225)$.

## 6.3 RESULTS ARE INDEPENDENT OF LEARNING DYNAMICS OF $g(\circ)$

To quantify the robustness of the results reported in the main body of the paper, we re-ran our experiments at different epochs during training of all 6 perceptual systems. Critically, the 270-th epoch for AlexNet was chosen to be displayed in the main body of the paper given that it is the approximate epoch after which the validation loss begins to diverge from the training loss for at least one of the perceptual systems. This promotes our choice of picking a relevant snapshot for AlexNet as two more points before and after the 270-th epoch, suggesting the 180-th (before 270) and 360-th epoch (after 270). The same effect (of a diverging validation loss of at least one system) can be seen for epoch 120 of ResNet18. Similarly, this encourages us to pick epoch 90 (before) and epoch 180 (after) as other reference snapshots to evaluate the interaction of learning dynamics with our pattern of results.

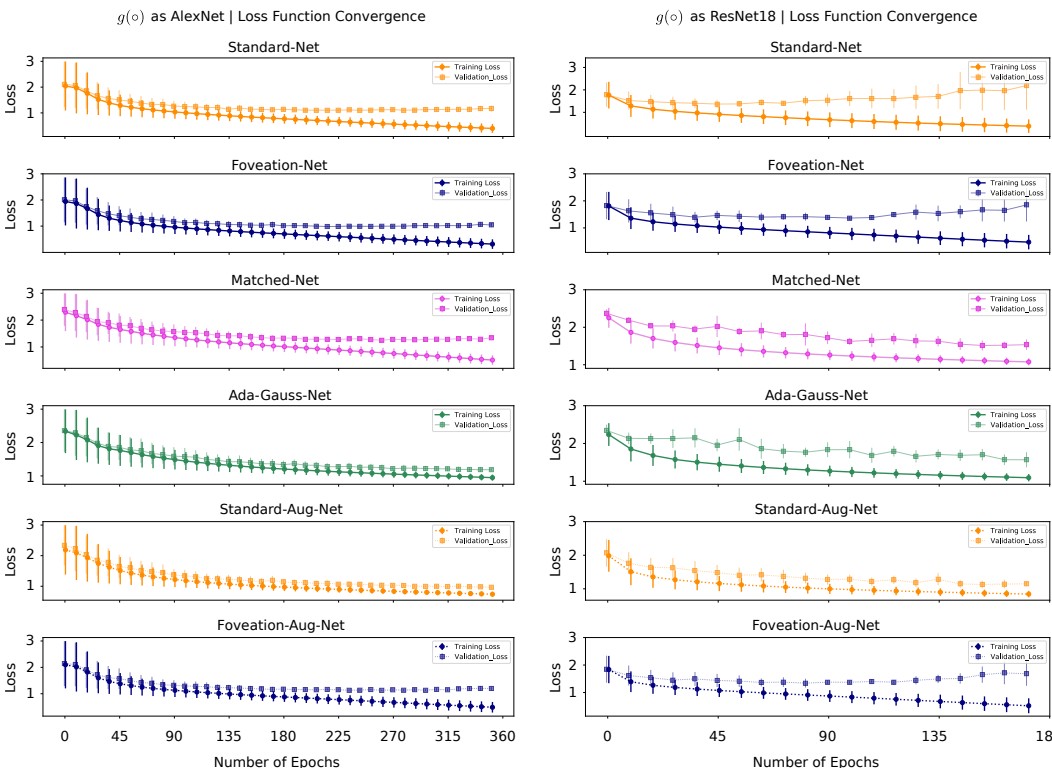

Figure 13: Learning: An averaged visualization of loss function convergence as a function of epochs for all 6 perceptual systems: Standard-Nets, Foveation-Nets, Matched-Nets, Ada-Gauss-Nets, Standard-Aug-Nets, Foveation-Aug-Nets. Each point in the plot is the average across the 10 different network runs from the locally averaged/smooth loss function per each 9 epochs. The epoch snapshots we show in our analysis are: AlexNet: 180, 270 (reported in main body of paper), 360; ResNet18: 90, 120, 180.

The collection of all the extended results for AlexNet + ResNet18 architectures at multiple epochs is shown in Section 6.6 (Generalization Extended), Section 6.7 (Robustness to Occlusion Extended), Section 6.8 (Window Cue-Conflict Extended) and Section 6.9 (Spatial Frequency Senstitivity Extended).

## 6.4 FOVEATION TRANSFORM SAMPLE VISUALIZATIONS

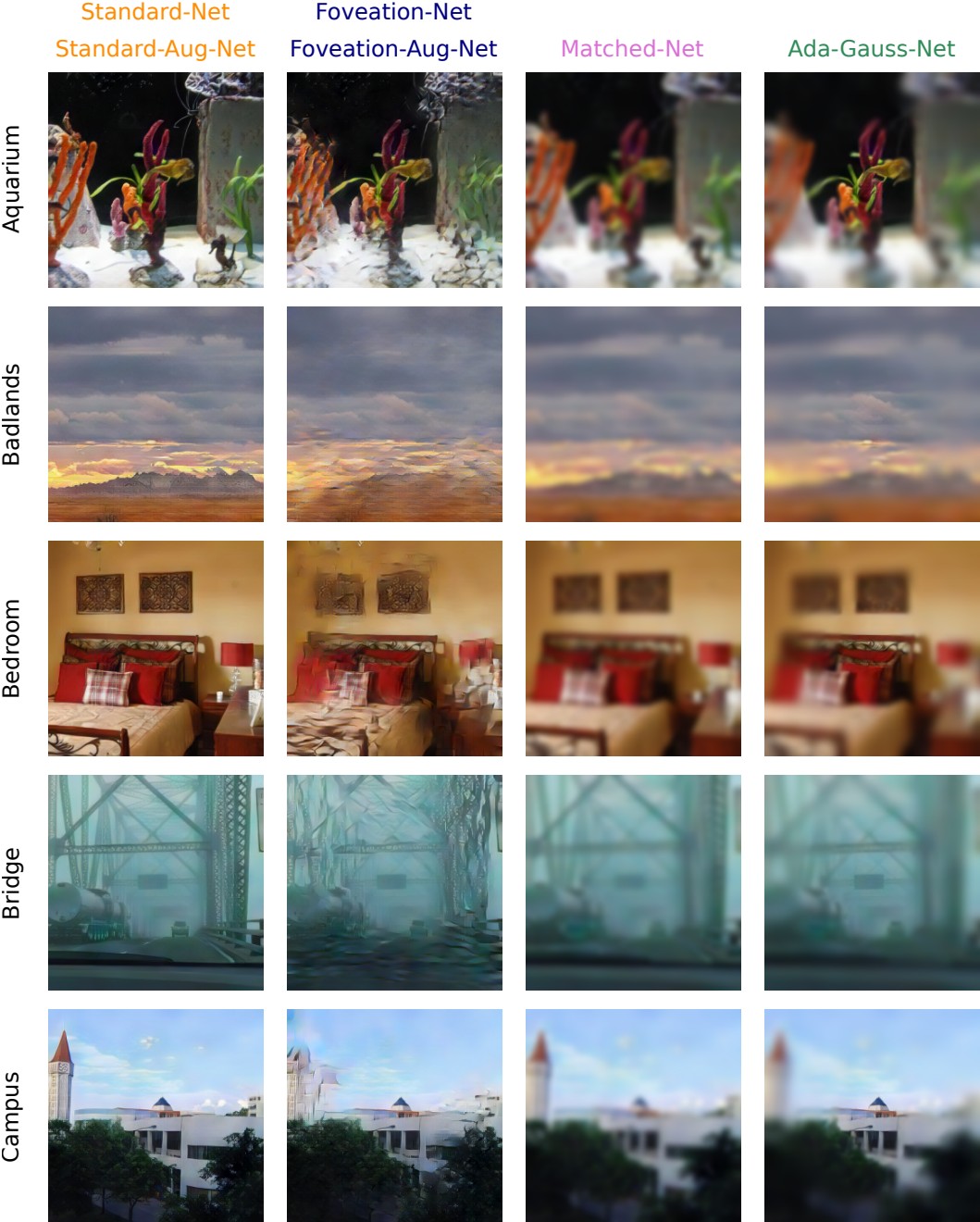

Figure 14: Sample Testing Image Mosaics (Part 1, not cherry picked).

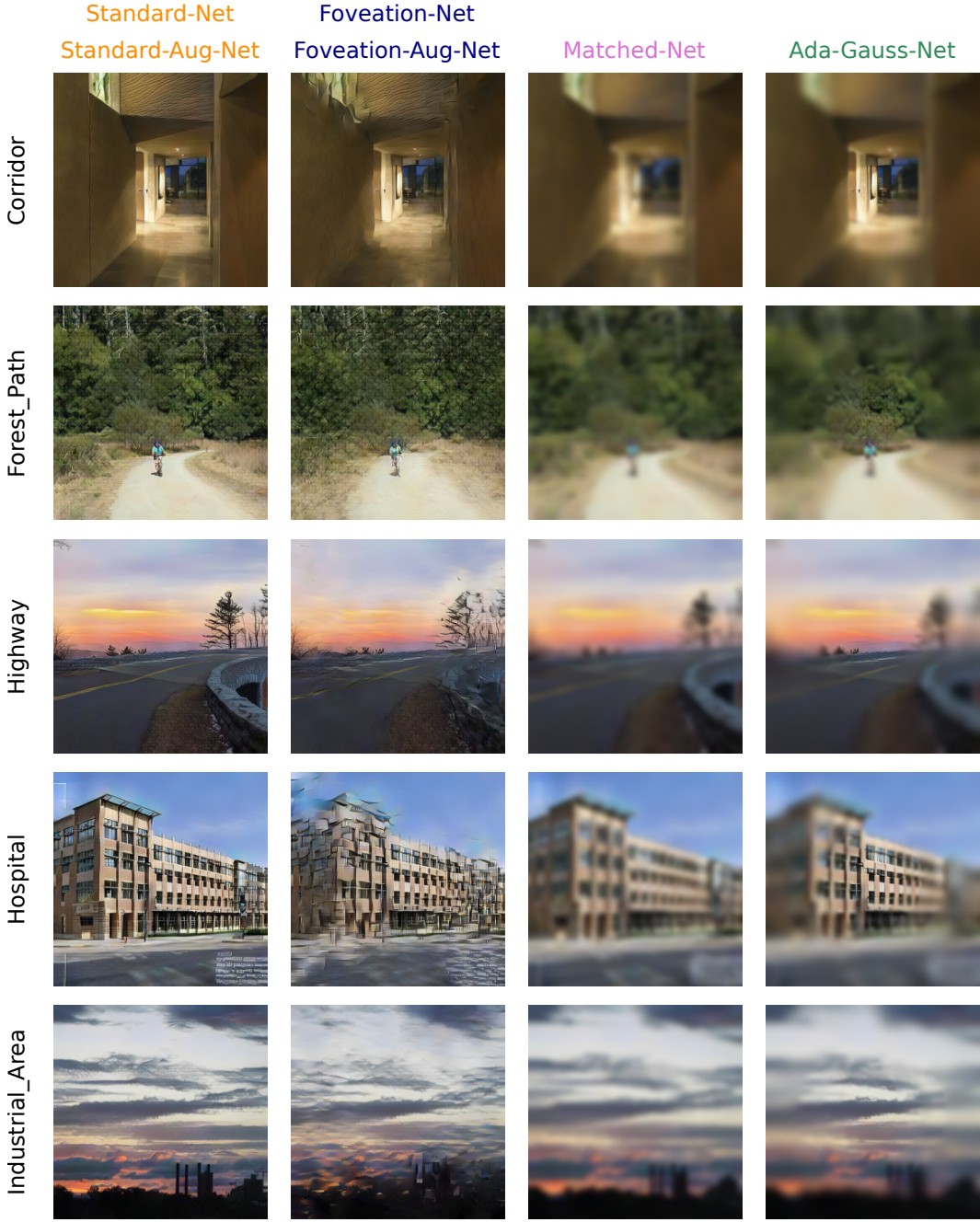

Figure 15: Sample Testing Image Mosaic (Part 2, not cherry picked).

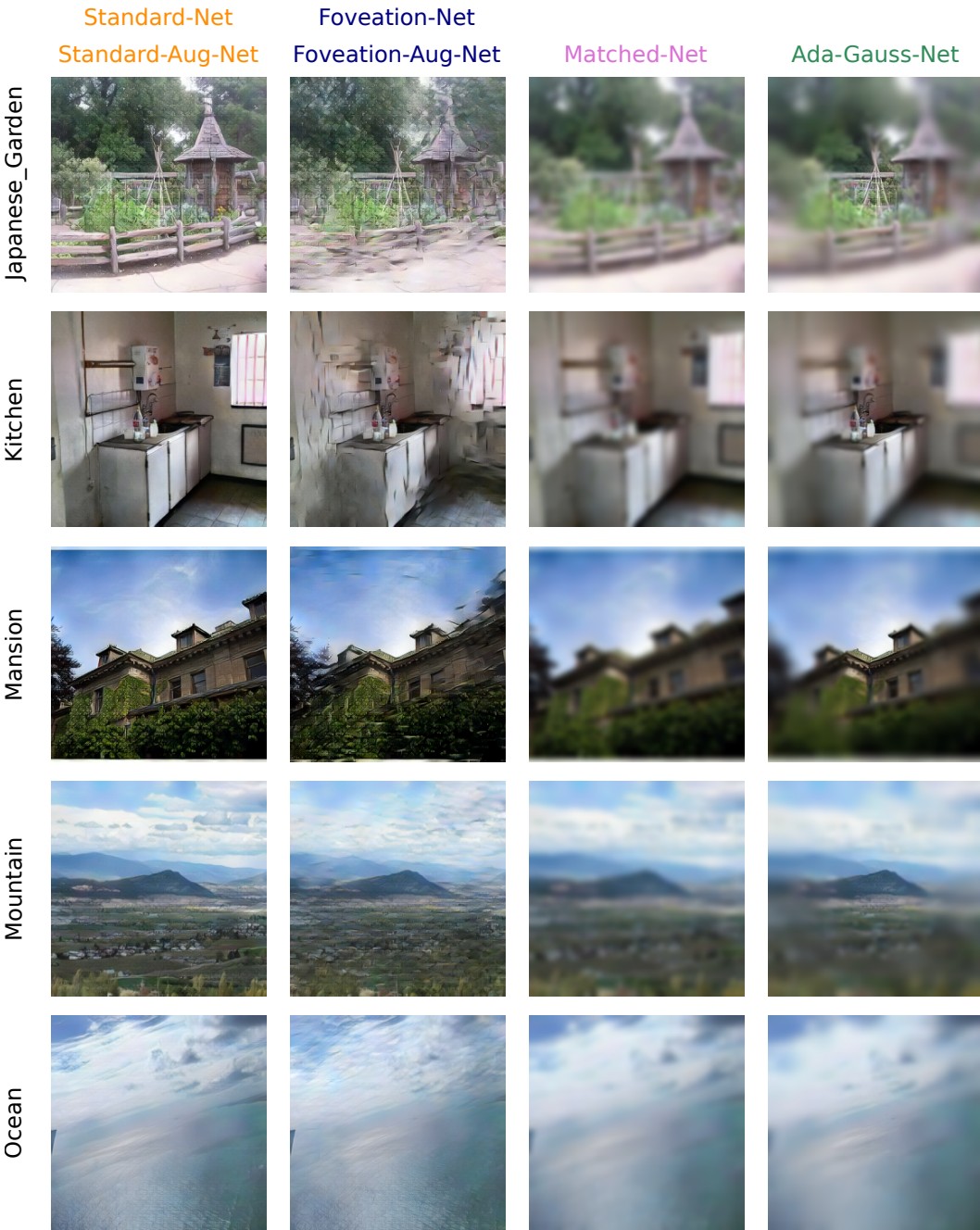

Figure 16: Sample Testing Image Mosaic (Part 3, not cherry picked).

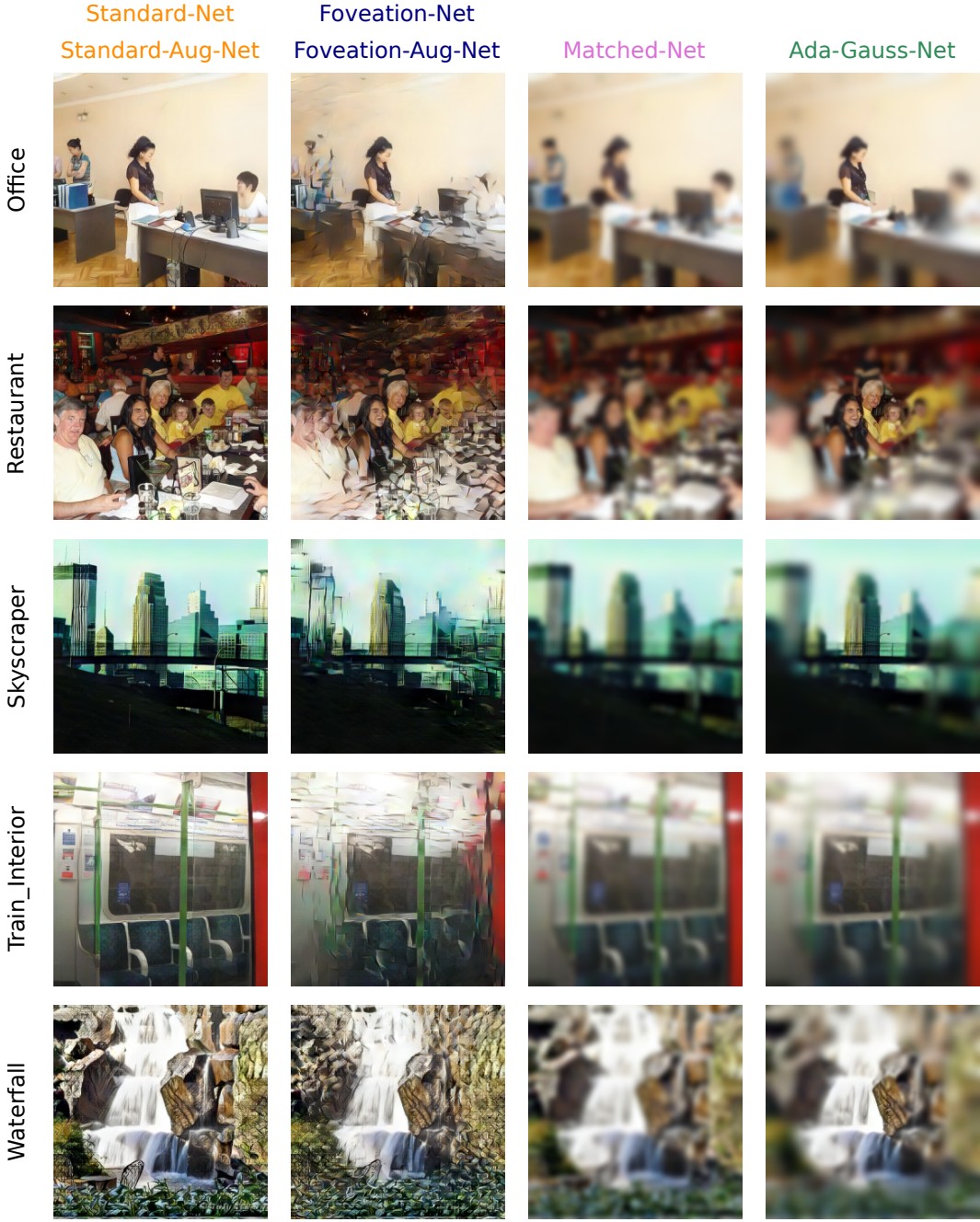

Figure 17: Sample Testing Image Mosaic (Part 4, not cherry picked).

### 6.5 DIFFERENCES ACROSS DIFFERENT FOVEATION MODELS AND RELEVANT WORK

There are currently 4 foveation models that implement texture-like computation in the peripheral field of view as shown in Table 1. We selected the Foveation Transform model given that it is computationally tractable to render a foveated image dataset (100'000) at a rate of 1 image/second (rather than hours Freeman & Simoncelli (2011) or minutes Wallis et al. (2017)). We did not use the highly accelerated model of Fridman et al. (2017) (order of miliseconds, that was based on the Texture-Tiling Model of Rosenholtz et al. (2012)) because it was: 1) Not psychophysically tested with human observers thus there is no guarantee of visual metamerism via the choice of texture statistics; 2) it does not provide an upper-bound computational baseline (similar to Standard-Net), when all the perturbations coefficients are off, given that the system is closed and lossy; 3) the code was/is unavailable at the time. However we think that re-running our experiments and testing them with other foveated models such as the before-mentioned is a direction of future work as we would be curious to see the replicability of our pattern of results across other texture-based peripheral models.

| Model | Freeman & Simoncelli (2011) | Wallis et al. (2019) | Fridman et al. (2017) | Deza et al. (2019) |
|---|---|---|---|---|
| Feed-Forward | - | - | ✓ | ✓ |
| Input | Noise | Noise | Image | Image |
| Multi-Resolution | ✓ | ✓ | - | - |
| Texture Statistics | Steerable Pyramid | VGG19 $conv$-$1_1, 2_1, 3_1, 4_1, 5_1$ | Steerable Pyramid | VGG19 $relu4_1$ |
| Style Transfer | Portilla & Simoncelli (2000) | Gatys et al. (2016) | Rosenholtz et al. (2012) | Huang & Belongie (2017) |
| Foveated Pooling | ✓ | ✓ | (Implicit via FCN) | ✓ |
| Decoder (trained on) | - | - | metamers/mongrels | images |
| Moveable Fovea | ✓ | ✓ | ✓ | ✓ |
| Use of Noise | Initialization | Initialization | - | Perturbation |
| Non-Deterministic | ✓ | ✓ | | ✓ |
| Direct Computable Inverse | - | - | (Implicit via FCN) | ✓ |
| Rendering Time | hours | minutes | miliseconds | seconds |
| Image type | scenes | scenes/texture | scenes | scenes |
| Critical Scaling ($vs$ Synth) | 0.46 | $\sim \{0.39/0.41\}$ | Not Required | 0.5 |
| Critical Scaling ($vs$ Reference) | Not Available | $\sim \{0.2/0.35\}$ | Not Required | 0.24 |
| Experimental design | ABX | Oddball | - | ABX |
| Reference Image in Exp. | Metamer | Original | - | Compressed via Decoder |
| Number of Images tested | 4 | 400 | - | 10 |
| Trials per observers | $\sim 1000$ | $\sim 1000$ | - | $\sim 3000$ |

Table 1: Metamer Model comparison. Redrawn from Deza et al. (2019).

There are several works that have used foveation to show a type of representational advantage over non-foveated systems. Mainly Pramod et al. (2018) with adaptive gaussian blur, and Wu et al. (2018) with scene gist, that have been targeted towards a computational goal in increasing object recognition performance. For scene recognition, only Wang & Cottrell (2017) has successfully modelled known behavioural results of Larson & Loschky (2009) via a dual-stream neural network that uses adaptive gaussian blurring and a log-polar transform. One key difference however is that we are interested in exploring the effects of *visual crowding* that is a phenomena linked to area V2 in the primate ventral stream (rather than retinal as in Wang & Cottrell (2017) which resembles our control condition: Ada-Gauss-Nets). In general, we are taking a complimentary approach where we *a priori do not know of a functional role of texture-based computation or prime ourselves to fit our model to a reference behavioural result*. Thus we explore what functionality it may have in comparison to a non-foveated system (Matched-Nets, Standard-Nets) or a foveated system that only implements adaptive gaussian blurring (Ada-Gauss-Nets). Table 2 highlights key similarities & differences between these papers and ours.

| Model | Wang & Cottrell (2017)) | Wu et al. (2018) | Pramod et al. (2018) | (Ours) |
|---|---|---|---|---|
| Image input type | scenes | objects | objects | scenes |
| Single/Dual Stream | Dual + Gating | Dual + Concatenation | Single | Single |
| Role of Single/Dual Stream | Coupling the fovea + periphery | Contextual modulation (scene gist) | Serializing the (single) two-stage model | |
| Foveated Transform (F.T.) | log-polar + adaptive gaussian blurring | Region Selection | adaptive gaussian blurring | Visual Metamer w/ texture-distortion |
| Stochastic F.T. | - | - | - | ✓ Deza et al. (2019) |
| Representational Stage of F.T. | retinal (Geisler & Perry, 1998) | "Overt Attention" | retinal (Geisler & Perry, 1998) | V2 (Freeman & Simoncelli, 2011) |
| Moveable Fovea | ✓ | ✓ | ✓ | ✓ |
| Explores the role of Eye-Movements | - | - | - | ✓ (Supplement) |
| Accounts for pooling regions | Implicit via adaptive gaussian blurring | - | Implicit via adaptive gaussian blurring | ✓ |
| Accounts for visual crowding | - | - | - | ✓ |
| Accounts for retinal eccentricity | ✓ | Implicit via cropping | ✓ | ✓ |
| Accounts for loss of visual acuity | ✓ | - | ✓ | Implicit via visual crowding |
| Critical Radius (Larson & Loschky, 2009) | 8 deg | Not Applicable (Objects) | | $\sim$ 8.9 deg (Estimated from Fig. 8) |
| Out of Distribution Generalization | - | - | - | ✓ |
| Robustness to Distortion Type | - | Blurring | Blurring | Occlusion |
| Spatial Frequency Preference | High (Fovea), Low (Periphery) | Low (Global) | High (Fovea), Low (Periphery) | High (Global) |
| Weighted Bias Emerges | Center/Fovea | Center/Fovea | Center/Fovea | Center/Fovea |
| Goal of Foveal-Peripheral Architecture | Fit Behavioural Results | Increase Recognition Accuracy | | Explore Perceptual Properties |
| Model System Focus | Human | Machine | Human | Hybrid |

Table 2: A summary set of Foveal-Peripheral CNN model characteristics.

## 6.6 GENERALIZATION (EXTENDED)

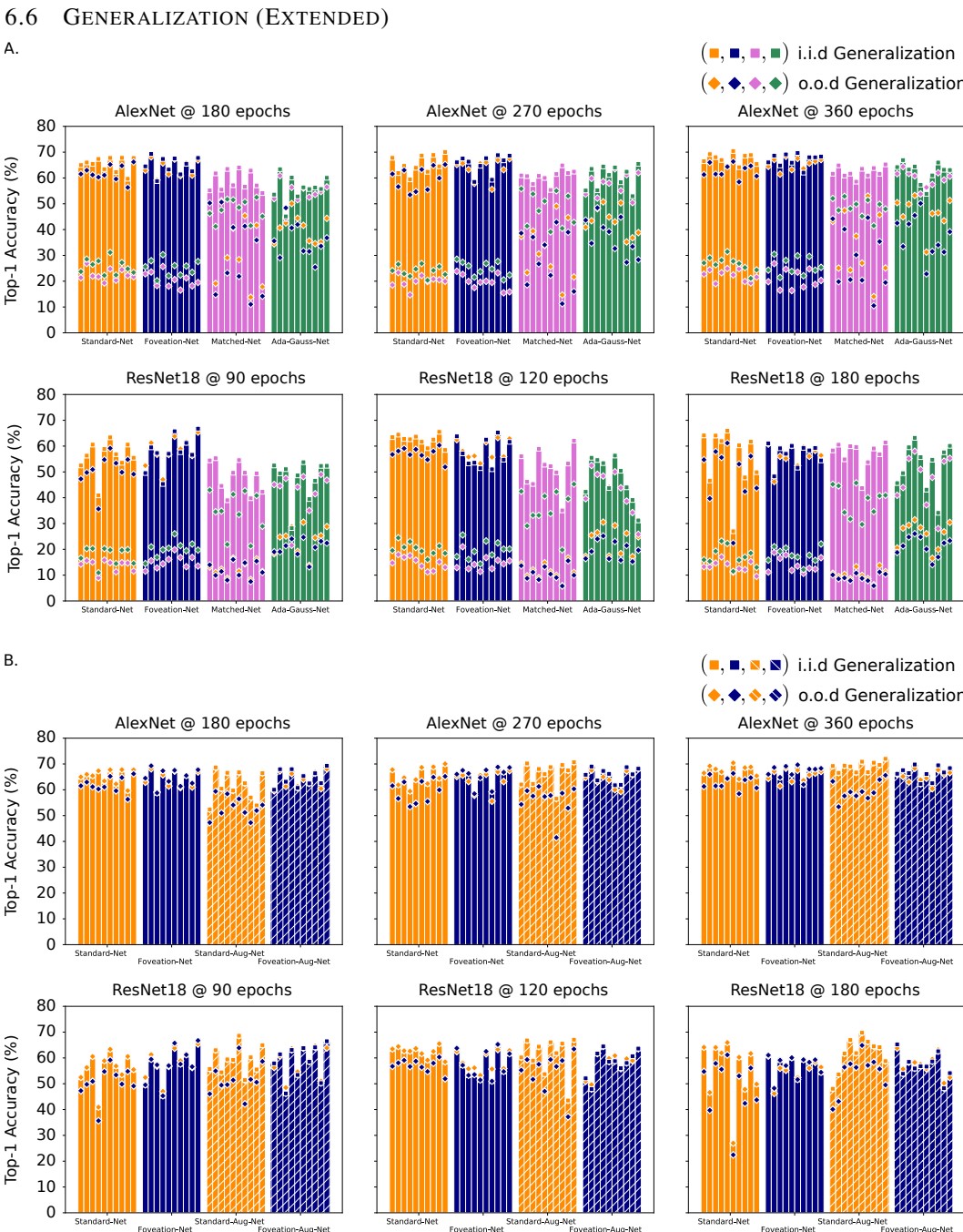

Figure 18: **Generalization:** The full i.i.d. Generalization and o.o.d Generalization plots for AlexNet and ResNet18 across multiple epochs of training. We observe that our results do not vary as a function of training epoch or network architecture: Foveation-Nets with greater i.i.d generalization across matched-resource systems, and Ada-Gauss-Nets with greater o.o.d generalization across all systems. This suggests spatially-varying computation provides a representational benefit in both the i.i.d and o.o.d setting contingent on the type of foveated computation (texture vs blur). Future work should evaluate combining both computations. A. Our 4 main perceptual systems: Standard-Net, Foveation-Net, Matched-Net, Ada-Gauss-Net; B. Standard-Net (re-plotted from above), Foveation-Net (re-plotted from above), Standard-Aug-Net (supplementary), Foveation-Aug-Net (supplementary). Notice that only gold and navy dots are shown for o.o.d plots because Standard-Aug-Nets were tested on Standard-Net inputs, and Foveation-Aug-Nets were tested on Foveation-Nets inputs.

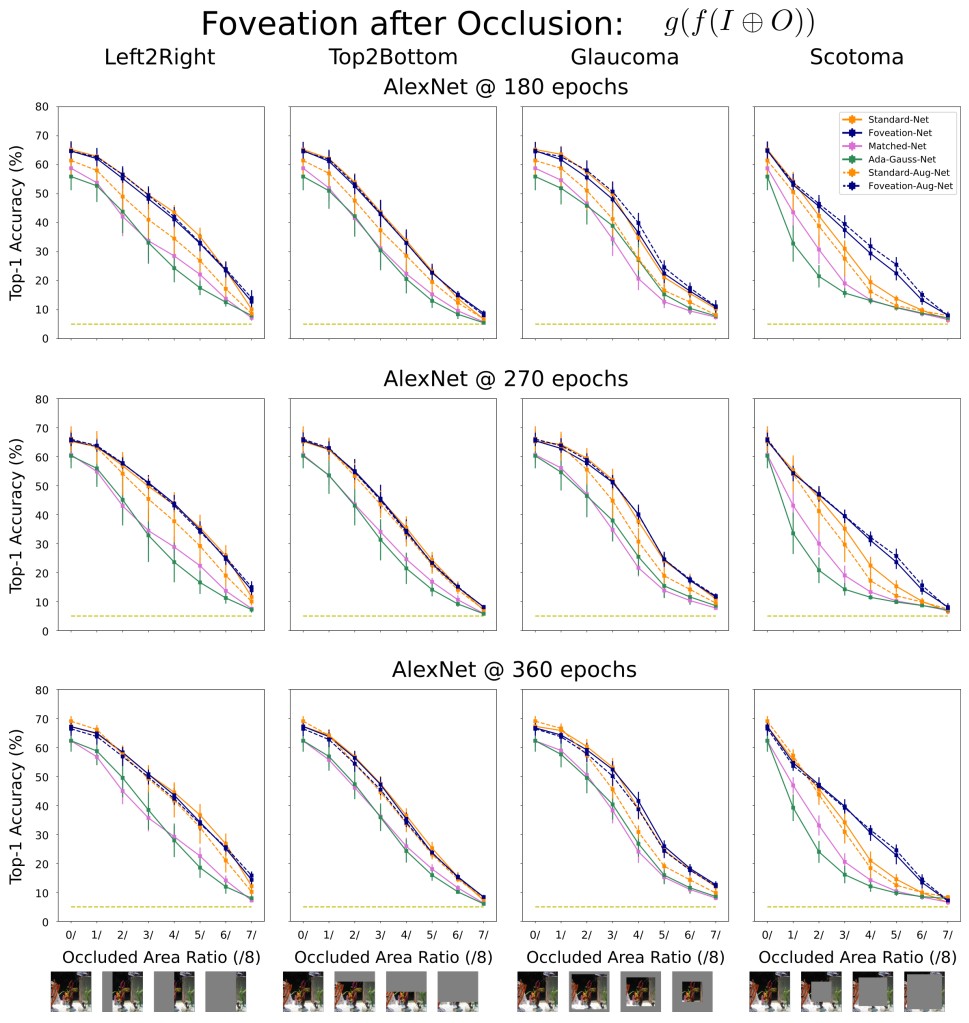

Figure 19: Robustness to Occlusion of All Perceptual Systems (Extended – Part 1/4).

## 6.7 ROBUSTNESS TO OCCLUSION (EXTENDED)

We decided to extend the robustness to occlusion analysis with 2 additional conditions: Left2Right and Top2Bottom, which are both occluding conditions where the stimuli is occluded with a gray patch from "left to right" and from "top to bottom" respectively. Altogether we find that the same pattern of results is sustained where Foveation-Nets are near the perceptual upper-bound provided by Standard-Nets. Both matched-resource systems: Matched-Net (spatially uniform and lower-resolution) and Ada-Gauss-Net (spatially varying with adaptive gaussian blurring) are *less robust to lateral, vertical, scotomal and peripheral occlusion* than Foveation-Net (spatially-varying with texture-based peripheral encoding) across both AlexNet and ResNet18 architectures as $g(\circ)$, and across different epochs (learning dynamics). Furthermore the reported findings in the main body of the paper with regards to the impact of the *visual crowding* mechanism (exclusively $f_*(\circ)$) as a filling-in operator, is sustained across both AlexNet and ResNet18 architecture as $g(\circ)$ and number of epochs (learning dynamics). These extended results are plotted in addition to the new controls Foveation-Aug-Nets and Standard-Aug-Nets in Figures 19,20,21,22. In these plots we once again show that the contribution of eye-movements in training (Foveation-Aug-Nets) provides marginal improvements in robustness to Foveation-Nets for both the pre and post foveation to occlusion conditions. On the hand, Standard-Aug-Nets show a pattern of results very similar to a combination of Matched-Nets and Standard-Nets. This should not be a surprise as Standard-Aug-Nets have been trained with multi-resolution images. Somewhat surprisingly, Standard-Aug-Nets *do not exceed the robustness* of Standard-Nets, showing a particularly counter-intuitive limitation to data-augmentation.

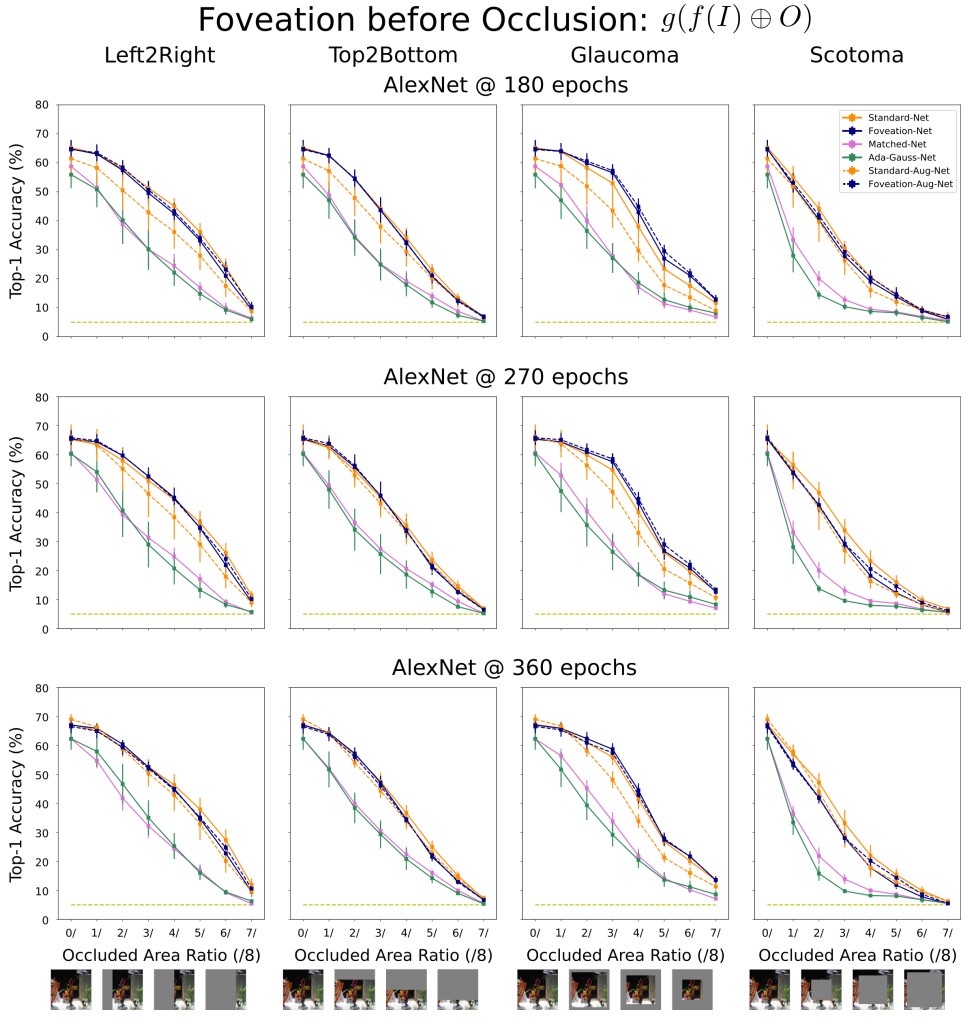

Figure 20: Robustness to Occlusion of All Perceptual Systems (Extended – Part 2/4).

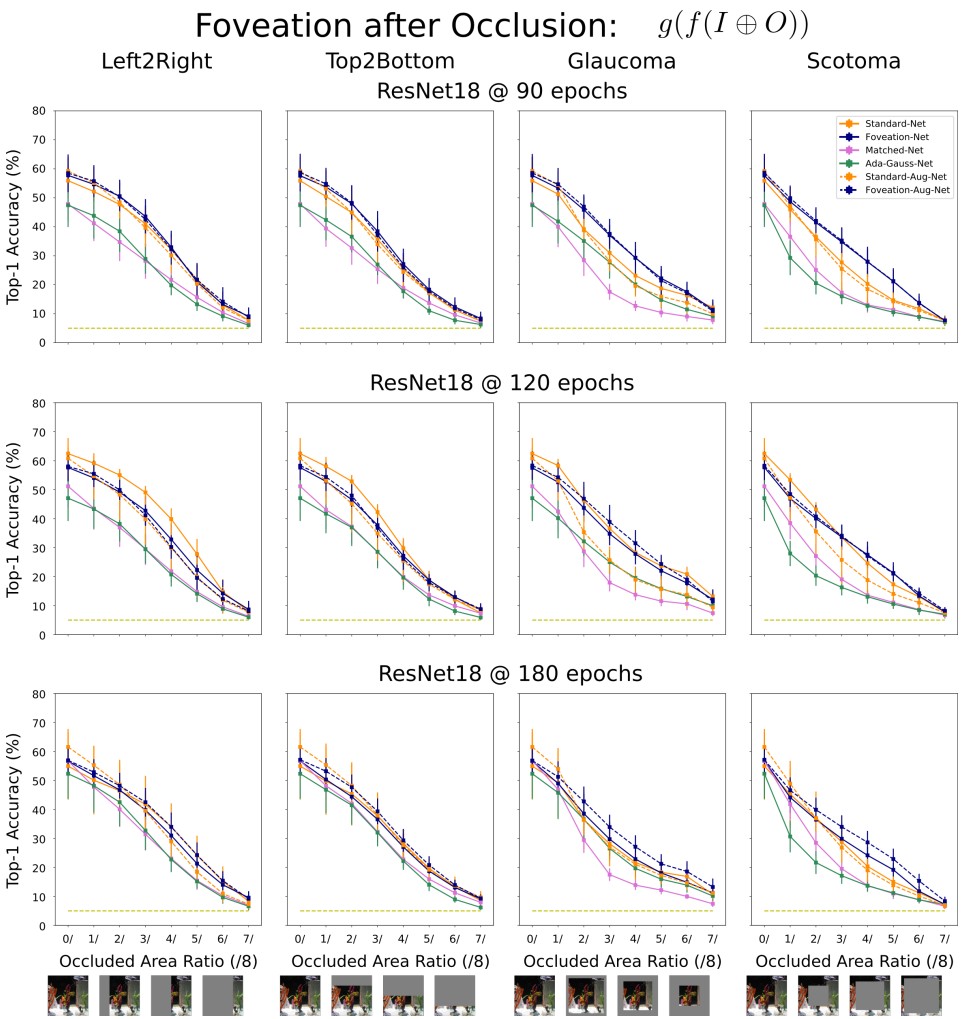

Figure 21: Robustness to Occlusion of All Perceptual Systems (Extended – Part 3/4).

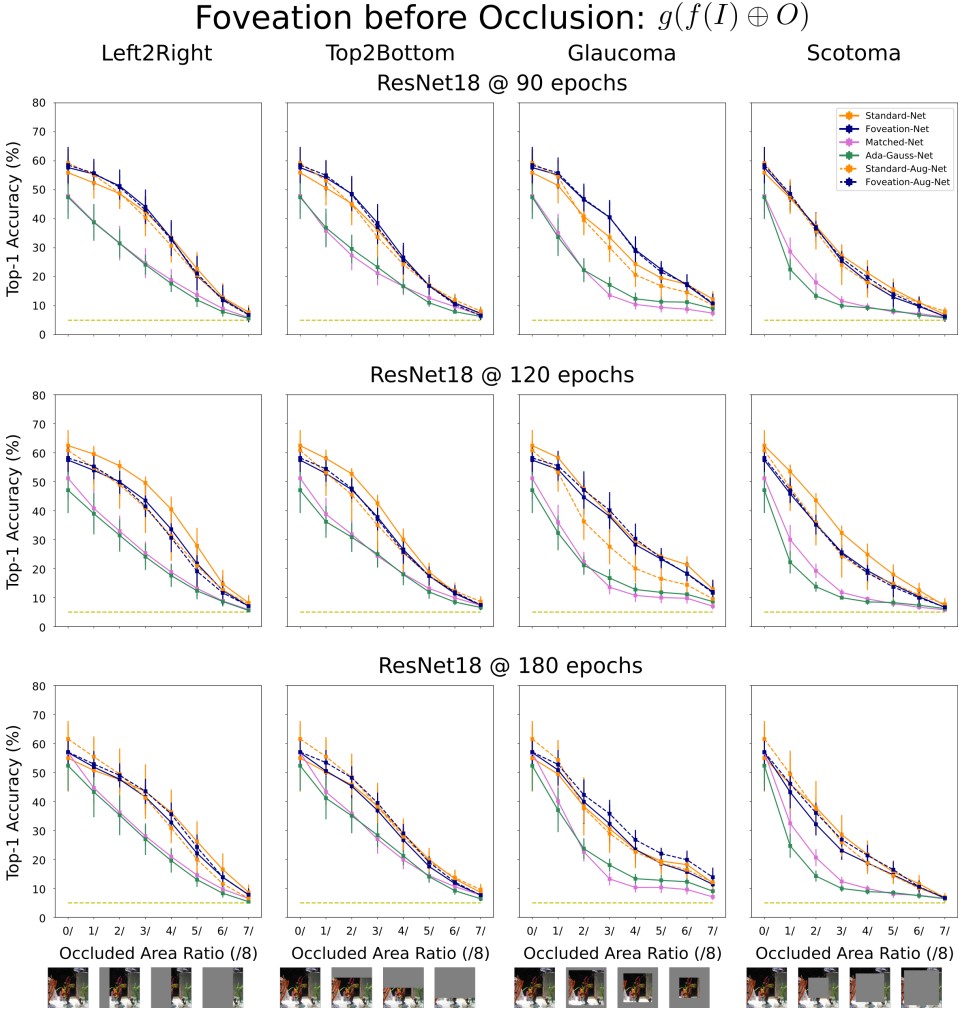

Figure 22: Robustness to Occlusion of All Perceptual Systems (Extended – Part 4/4).

## 6.8 WINDOW CUE-CONFLICT (EXTENDED)

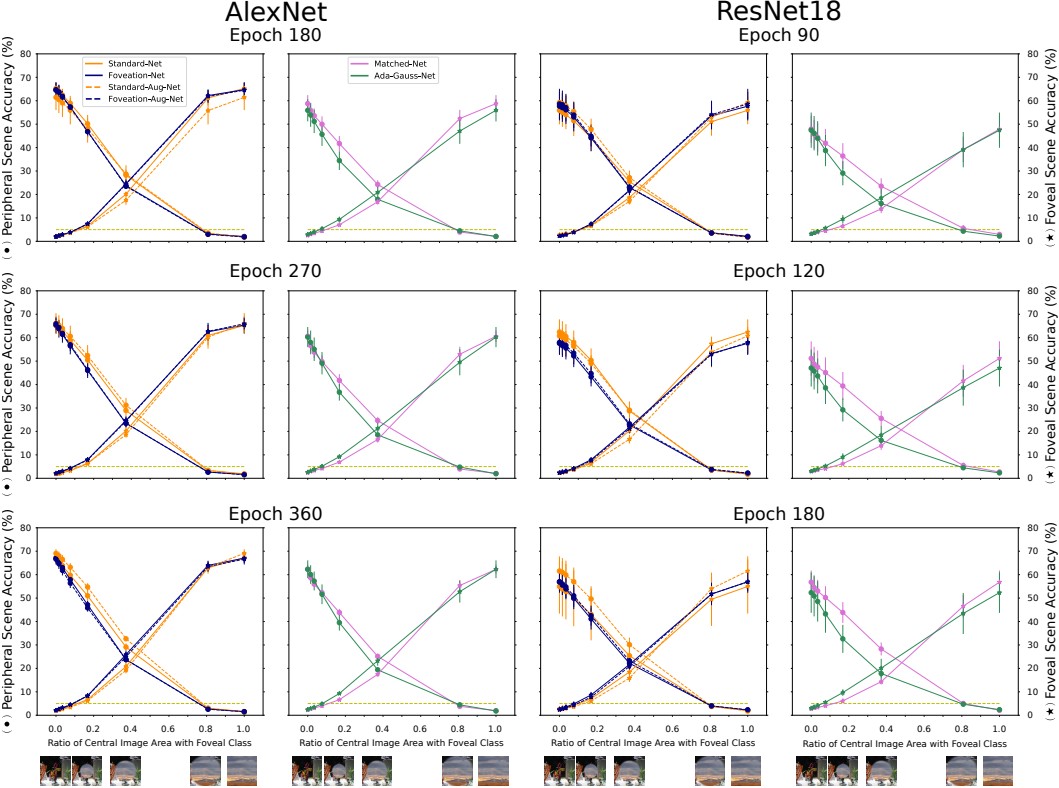

Figure 23: Window Cue Conflict Experiment: The pattern or results with regards to a greater foveal bias for any *foveated perceptual system* over *non-foveated perceptual systems* remains independent of the network architecture ($g(\circ)$) and the epoch. This can be verified by finding the cross-over points for Foveation-Nets, Ada-Gauss-Nets, and Foveation-Aug-Nets being placed more leftwards than Standard-Nets, Matched-Nets and Standard-Aug-Nets. These results are independent of potential perceptual differences at testing time *i.e.* baseline. For example, see AlexNet @ 360 epochs, or ResNet18 @ 120 or 180 epochs, where the cross-over point for Standard-Nets, Matched-Nets and Standard-Aug-Nets is still shifted more biased towards the right than Foveation-Nets, Ada-Gauss-Nets and Foveation-Aug-Nets – implying a greated need for foveal area to arrive to the point of subjective equality (PSE). A final note on the interpretability of these results is that this foveal bias is being tested *after* the foveation transforms are computed (similar to our post-foveation occlusion experiments), such that no changes in area are driving the revealed biases, and thus the bias is driven purely by the learned representation.

## 6.9 SPATIAL FREQUENCY SENSITIVITY (EXTENDED)

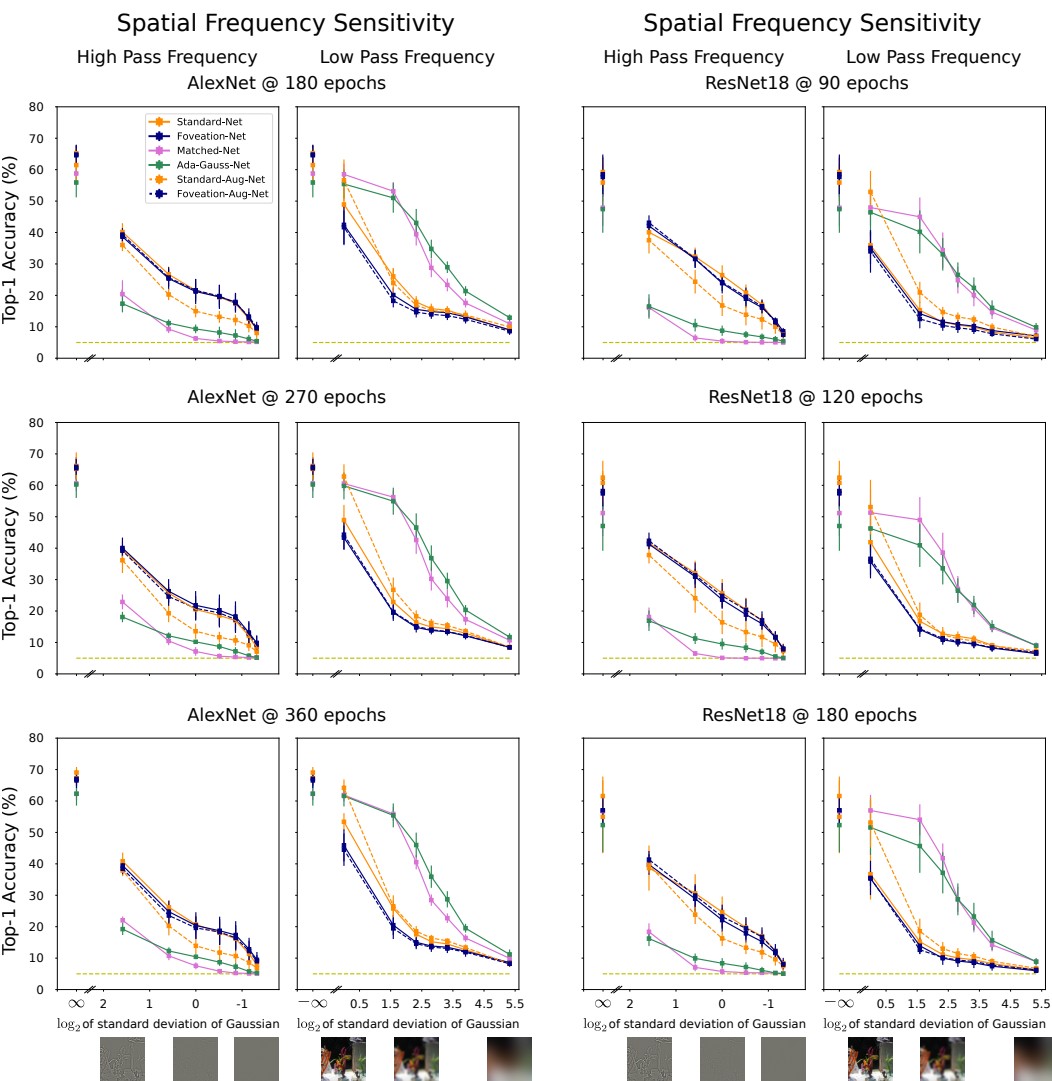

Figure 24: Spatial Frequency Sensitivity for AlexNet and ResNet18 across multiple training epochs: There are no notorious differences for high pass frequency sensitivity across network architecture and epochs in comparison to the results reported in the main body of the paper. Specifically, these patterns are: no differences between Foveation-Nets, Foveation-Aug-Nets and Standard-Nets (all three statistically tested with paired t-tests against each other, n.s.). There are differences between Standard-Nets and Standard-Aug-Nets (a greater bias to high pass spatial frequency sensitivity in Standard-Nets). These 4 systems are notably also more biased to high pass spatial frequency than Ada-Gauss-Net and Matched-Net in an orderly fashion. The opposite pattern of results hold for low pass frequency sensitivity.

Images shown to each network were the respective images from each training-testing distribution pair. Foveation-Nets were shown foveated images with a center fixation, Standard-Net were shown non-foveated images, Matched-Nets were shown matched-resource non-foveated (yet blurred) images, and Ada-Gauss-Nets were shown foveated images with adaptive gaussian blurring. Thus, both High Pass Spatial Frequency and Low Pass Spatial Frequency experiments were conducted at the *post-foveation* stage to directly examined the learned representations of $g(\circ)$: the second stage of each perceptual system.

The size of all shown images was $256 \times 256 \times 3$, thus the units of the gaussian filters specified from Section 4.4 are in pixels. For a given Gaussian filtering operation $\mathcal{G}_\sigma$ for a given standard deviation $\sigma$, low pass spatial frequency (LF) images were computed via:

$$LF(I^C) = \mathcal{G}_\sigma(I^C) \tag{2}$$

for each channel $C$. In the main body of the paper we used the summarized notation $\mathcal{F}_L(\circ)$ for low-pass frequency filtering operator. Similarly, High Pass Spatial Frequency (HF) image stimuli were computed via:

$$HF(I^C) = I^C - \mathcal{G}_\sigma(I^C) + \text{mean}_{\text{val}}^C \tag{3}$$

where $\text{mean}_{\text{val}}^C$ (which we call the residual in the main body of the paper) is the average of image intensity over the held-out validation set for each channel $C$, a small extension from Geirhos et al. (2019) as our image stimuli is in color vs grayscale. In the main body of the paper we used the summarized notation $\mathcal{F}_H(\circ)$ for the high pass frequency filtering operator.

