# OpenReview forum: "Emergent Properties of Foveated Perceptual Systems"
_ICLR.cc/2021/Conference — Reject_

### Official Review · AnonReviewer3 · 2020-10-27
**The paper provides a comprehensive study to evaluate how a texture-based foveated inductive bias influences visual perception from several perspectives.**

**Rating:** 7
**Confidence:** 3

**Review:**

The authors compared Foveation-Net with three other stage-one visual systems: Standard-Net with non-foveated transform; Ada-Gauss-Net as a matched-resource control for Foveation-Net (to test the contribution of the foveation transformation comparing to other spatially-varying processing); Matched-Net as a matched-resource control for Standard-Net. Specifically, the authors did four sets of experiments on these visual systems to evaluate their generalizability, robustness, image-region bias, and spatial frequency sensitivity.

Overall, the methods are solid and clearly stated. The results suggest some perceptual advantages and specific representational properties yielded by texture-based foveation transformation.

##### Specific comments and questions:
1. It is not clearly stated how to read the plot in Fig. 5B. And why using a 20-way scene categorization task to evaluate the model performance?
2. The results in experiment 3 suggest Foveation-Net and Ada-Gauss-Net has higher foveal bias. Is this simply because the effective sampling density is higher in the image center (as suggested by Fig. 5A) for these two visual systems? Then are these two visual systems more computationally efficient in general?
3. In section 3.4, the authors argued that the Foveation-Net may enforce a shape-bias since "these Spatial Frequency curves show similar trend as SIN". Are these curves refer to the ones in Fig. 8B? But by visually inspecting this figure, it seems the trend of Standard-Net is quite closed to Foveation-Net. Please let me know if I misunderstood this argument.
4. What would be the effect on model performance if the fixation is not in the center for Foveation-Net and Ada-Gauss-Net?

---

> ### Author Response · Authors · 2020-11-19
> **Comments to AnonReviewer3 [1/2]**
>
> Thank you for your interest R3 and your follow-up questions!
>
> - “In section 3.4, the authors argued that the Foveation-Net may enforce a shape-bias since "these Spatial Frequency curves show similar trend as SIN". Are these curves refer to the ones in Fig. 8B? But by visually inspecting this figure, it seems the trend of Standard-Net is quite closed to Foveation-Net. Please let me know if I misunderstood this argument.”
>
> You are correct, though there are two parts to this observation: 1) The real (and fair) comparison should be with regards to Matched-Net and Ada-Gauss-Nets (that is also foveated), as these two systems are perceptually resource matched to Foveation-Nets. Standard-Net here only provides a “perceptual upper bound” as the perceptual resources are unmatched of this transform to all others. We will make this clear as we continually update our paper throughout the open discussion time window; 2) Foveation-Nets is still showing a marginal improvement of high spatial frequency sensitivity (and opposite for low spatial frequencies) than Standard-Nets for *scenes*. We do not make any strong claims about a shape bias, but only suggest this as a form of a conjecture since the shape bias of Geirhos et al. 2019 and recently with more detail in Hermann et al. 2020; was tested on a different dataset (ImageNet) and thus had a different task which was object classification. It is well known in vision science that scene and object recognition are two fundamentally different problems in both human and machine vision; ex: human classify objects through shape-based classification strategies, while scenes are classified via texture-based strategies in humans. Further, objects have an intrinsic shape/outline/contour, it is debatable to say the same about scenes (they have a `gist’ see Oliva & Torralba, IJCV 2001).
>
> Further we’d like to emphasize a non-trivial observation: Geirhos et al. 2019 (Texture Bias), overcame the texture-bias in deep networks by training networks on style-transferred images (texture distorted images) with Adaptive Instance Normalization. The Deza et al. ICLR 2019 Metamer model that we used also uses Adaptive Instance Normalization (Huang & Belongie ICCV 2017) locally at each receptive field level. In a way the Deza et al. Metamer model is an “eccentricity dependent localized auto-style transfer” model and, it could be that these texture-like computations in the periphery act as an implicit proxy/regularizer that enforces a shape bias (highly correlated with high spatial frequency bias) in machines -- and possibly humans. It is also not obvious how one would test for a shape bias in *scenes* (if classification strategies are driven by texture or other cues in contrast to objects!).
>
> This is a far-stretched idea, and an open question (hence our initial conjecture that was preliminary based on scenes), but we think it is a seed worth planting as it is a subject of on-going work for the CV/ML community as there has been no solution to finding a biologically plausible training objective or transformation in deep neural networks that imposes a shape bias (*for objects*), to the same degree as humans (including self-supervised objectives as in Geirhos et al. SVRHM 2020). It is an exciting direction that we will articulate more in the updated version of our manuscript as we continue to receive your feedback as this is a direction we are working on as well!
>
> - “What would be the effect on model performance if the fixation is not in the center for Foveation-Net and Ada-Gauss-Net?”
>
> This is a great question, we did indeed explore this in our original submission: We trained a system that performed multiple eye-movements in the scene during training (Foveation-Aug-Net), and found that this did not produce any statistically significant differences compared to Foveation-Net, though there was a slight tendency to do marginally better. We think that this is primarily due to the fact that eye-movements do not boost performance in scene recognition (compared to object recognition; See Akbas & Eckstein, PLOS Computational Biology 2017: “Foveated Object Detector”).  In the extra page of our manuscript (now page 5), we have very briefly made mention of Foveation-Aug-Net for readers who would like to see the effects of the role of eye-movements for our foveated models with regards to scene categorization in the Supplementary Material.

---

> > ### Author Response · Authors · 2020-11-19
> > **Comments to AnonReview3 [2/2]**
> >
> > - “It is not clearly stated how to read the plot in Fig. 5B. And why using a 20-way scene categorization task to evaluate the model performance?”
> >
> > Thanks R3: Fig5B: The squares in the bar graphs represent the i.i.d generalization performance while the diamonds o.o.d generalization performance across each one of the different network runs for each perceptual systems (hence 10 bars per system). Please see our updated manuscript with page 5 that should hopefully now make (Figure 6B; previously 5B) more accessible. But if it is still hard to decipher, please let us now! To your second point, we performed 20 way scene categorization for 2 main reasons: 1) It can give us a better learning signal for the networks, such that further analysis with regards to robustness/generalization can be driven by the complexity of the perturbation rather than the task (suppose training/testing on 1000-way scene classification, a small perturbation would likely make each system perform poorly just by the nature of having many labels rather than the actual perturbation). Moreover, work of Geirhos et al. 2018 indeed showed several key results on comparing generalization of humans and machines with only 16 object classes, thus we thought 20 classes for scenes seemed reasonable as well (we report Top-1 accuracy). 2) We needed to train each scene with the same amount of high-resolution images (4500 per class; originally at 512x512),to prevent class imbalance and also because it made it computationally tractable to train the family of 80 networks on a subset of the Places dataset (100k) images rather than the whole dataset (2 million), plus the computational cost of rendering each foveated images (another potential 6 million images to render, rather than 400k which is more computationally tractable and easier to distribute as a dataset).
> >
> > We’ve added a new figure in page 5 and a summarized Methods section in our updated manuscript that elaborates more on the dataset. Let us know if this has made things more clear.
> >
> > - The results in experiment 3 suggest Foveation-Net and Ada-Gauss-Net has higher foveal bias. Is this simply because the effective sampling density is higher in the image center (as suggested by Fig. 5A) for these two visual systems? Then are these two visual systems more computationally efficient in general?
> >
> > This is a great follow-up question R3! Actually the center(foveal) image bias is being tested and derived when stimuli is shown *after* the foveation transform to remove any confounding effects of the crowding operator that fills in the image (as seen in the occlusion results). The foveal image bias presumably rises because the system has learned to weigh the center region of the image more strongly than the warped periphery (thus removing any type of uniform weighing as observed for the non-foveated systems: Standard-Net and Matched-Net). It *can not* be attributed just by virtue of having a high sampling density in the fovea as Standard-Net (that is non-foveated) has the same amount of *foveal* perceptual resources than Foveation-Net and Ada-Gauss-Net and yet presents no foveal bias as strong as Foveation-Net and Ada-Gauss-Net. Recall that Standard-Net does indeed have overall *more* perceptual resources than all other systems, and even then does not present a center/foveal bias (Figure 8 C). To your follow-up question, one could maybe make an argument about efficiency but only in terms of `'re-distributing' the resources through the learned weights as the three systems (Foveation-Net, Matched-Net and Ada-Gauss-Net) are perceptually resourced matched via the Rate-Distortion Optimization we did in Section 2.3 -- and explained more thoroughly in the Supplement: Figure 10 & 11.

---

### Official Review · AnonReviewer1 · 2020-10-29
**Not sure if the premise of the paper is well-conceived**

**Rating:** 3
**Confidence:** 4

**Review:**

# Update after discussion
The discussion has reinforced my concerns. I cannot fully follow the logic of the paper and am not convinced the controls are useful. Therefore  I stand by my original assessment.


# Original Review

The paper proposes to use texturization of the periphery in images as a proprocessing step for scene classification and claims "greater iid generalization, high spatial frequency sensitivity and robustness to occlusion" for such models. The model is essentially a concatenation of the synthesis procedure for metameric images by Deza and colleagues (2019) followed by AlexNet as a classification head.


Strengths:
+ Tries to address an interesting question: why does human vision treat the periphery mainly as texture?
+ In-depth investigation of the model and three controls


Weaknesses:
- Unclear if the premise of the paper is well-conceived
- No evidence provided that "foveated" images are actually metameric
- "Matched-resource" controls are somewhat questionable
- Definition of "foveation" seems strange


Details on major issues (all of which would need to be addressed in a convincing manner for my score to change):

My main issue with the paper is that I don't understand its logic. The results by Rosenholtz, Simoncelli, Wallis and others show that some information about the image is discarded in human peripheral vision and, to some extent, peripheral vision "cares" only about the texture statistics of an image rather than the detailed composition. The hypothesis is that this is the same effect that has been described as crowding.

Now, the paper seems to put forward the hypothesis that distorting (texturizing) images in a way that they remain indistinguishable (metameric) for humans somehow enhances their discriminability. I don't understand what's the reasoning behind this logic. Why would throwing away information and distorting an image be useful? The original image is by definition part of the equivalence class of metameric, so why should a distorted image somehow be easier to classify by a neural network than the original image? Both images are perceived in the same way by humans. It seems like a strange proposal to me, but it's possible that I'm missing something. I'd like the authors to explain their reasoning better.

Having said that, even if we put these fundamental concerns aside, there are a number of practical issues:

(1) The presented images don't appear metameric to me, and the paper does not provide any psychophysical data showing that they are. For instance, for the three images labeled "Metameric" in Fig. 5A, I have a very hard time believing that they are metameric to human observers. Could the authors provide psychophysical evidence that this is the case?

(2) I am not sure about the purpose of the "matched-resource" control and why SSIM is the right metric to use here. If it's about information content in the image, I think it should be evaluated in terms of mutual information between the image and the labels (which is hard); if it's about information available to human observers, then a perceptual metric is the right approach, but SSIM is a very poor one that doesn't correspond well with human perception. Therefore, I am skeptical that these baselines are really useful.

(3) The claim “greater iid generalization, high spatial frequency sensitivity and robustness to occlusion emerged exclusively in our foveated texture-based models” seems to be an overstatement. There appears to be basically no difference between foveated and standard net. In all Figures (5–8) the differences are mostly within the range of 1 SD and the order of the two models changes sometimes. The only effect that might be significant is the difference in the scotoma condition for intermediate to high occlusion conditions. The only robust difference is between standard+foveation and MatchedNet+AdaGauss, where the latter seem to carry much less information (see points (1) and (2) above).



Minor comments

- Sec 3.4: The connection between Gaussian blur and scale invariance is not clear
- Sec 3.4: The following statement is trivial, since the images were low-pass filtered: “We found that Foveation-Nets and Standard-Nets were more sensitive to High Pass Frequency information, while Ada-Gauss-Nets and Matched-Nets were sensitive to Low Pass Frequency stimuli”
- Fig 8: It is unclear how it argues for shape bias.

---

> ### Author Response · Authors · 2020-11-19
> **Comments to AnonReviewer1 [1/4]**
>
> Thank you for your feedback R1! We’d like to address several points that seem to may have caused confusion compared to the excitement of other reviewers who are more than welcome to read/reply to this rebuttal as well if something is not clear. This is a great opportunity to clear some doubts as several of these questions have arised during informal and productive discussions with colleagues.
>
> - “Now, the paper seems to put forward the hypothesis that distorting (texturizing) images in a way that they remain indistinguishable (metameric) for humans somehow enhances their discriminability. I don't understand what's the reasoning behind this logic. Why would throwing away information and distorting an image be useful? The original image is by definition part of the equivalence class of metameric, so why should a distorted image somehow be easier to classify by a neural network than the original image? Both images are perceived in the same way by humans. It seems like a strange proposal to me, but it's possible that I'm missing something. I'd like the authors to explain their reasoning better.”
>
> We’d love to have the chance to especially clarify this point: >>> “Why would throwing away information and distorting an image be useful?” One could similarly ask “Why would training networks on images with altered textures perform better (robustness) than training on non-textured altered images?” (Texture Bias; Geirhos et al., Oral ICLR 2019); “Why would training networks on distorted ‘robust’ adversarial images perform better (robustness) than training on regular natural images?” (Adversarial Examples Are Not Bugs, They Are Features; Ilyas et al., Spotlight NeurIPS 2019); “Why would having a V1-based front-end visual processor with noise beat a fully differentiable end-to-end-trained deterministic system in terms of adversarial robustness?” (Dapello, Marques et al.; Spotlight NeurIPS 2020); and pushing this argument even further “Why would filtering an image aid in performance?” (Pramod, Kitti & Arun. ArXiv 2018 -- who prematurely showed that adaptive low-pass filtering can reduce false alarm rates for a foveated *object* detector).
>
> The answer to these questions (that in hindsight seem obvious) is because these counter-intuitive inquiries oftentimes shed light into the role of computational mechanisms that we thought we understood -- indeed, we are not making a wild guess, as we do have a biological prior which is that humans perceive and *evolved* with a spatially-varying (foveated) visual field and that texture-like representations are in fact used in the visual periphery (Balas, Nakano & Rosenholtz. Journal of Vision. 2009) -- and we will only find out if any *representational* benefits for machines (or humans) in terms of generalization, robustness and other biases, are attainable until we conduct the physical, or in this case computational, experiment. We do not necessarily hypothesize the foveated system to do better in accuracy (though this idea is not far-fetched if one conceives texture-based distortions as an implicit regularizer), but rather hypothesize that the system will be *different* in general (hence using a Menu of different perceptual tasks to probe these perceptual differences); and if so, (1) by how much?; and (2) are these positive or detrimental differences for each system. In particular the emphasis here (as we will reinforce later in the rebuttal [Part 3]) is that the comparisons should be focused on Foveation-Net vs Matched-Net & Ada-Gauss-Net. Standard-Net provides a perceptual upper bound that is not comparable to other systems, but is still relevant to report and analyze.
>
> Finally, perhaps we have missed this, but we never hypothesized in the main body of the paper (or Introduction) that “(texturizing) images in the way that remain metameric for humans somehow *enhances their discriminability*”. We do however argue as we mentioned before and in the last 2 paragraphs of our Introduction, that this texture-based transform could potentially provide a *functional benefit* (though this does not need to be directly tied to performance; it could be robustness or other perceptual biases as we explore in our paper).

---

> > ### Author Response · Authors · 2020-11-19
> > **Comments to AnonReviewer1 [2/4]**
> >
> > - R1: “(1) The presented images don't appear metameric to me, and the paper does not provide any psychophysical data showing that they are. For instance, for the three images labeled "Metameric" in Fig. 5A, I have a very hard time believing that they are metameric to human observers. Could the authors provide psychophysical evidence that this is the case?”
> >
> > You are absolutely correct, the scaling factor has been exaggerated in our experiments (to 0.4, instead of 0.25) to force the deep neural network to engage in picking up these distortions under the same type of texture-based distortion that resembles crowding with a stronger parametrization. One may argue that not using the scaling factor of 0.25 is a bad experimental choice (“Why did you not keep it metameric as for humans!?”), but here our goal is to show the implications of crowding based transforms in the visual periphery of a *machine* -- where even the notion of `degrees in the visual field’ is fuzzy and allows us to play with stronger peripheral distortions. In other words we’d like to study what is the impact of a (strong) texturized periphery on a model's ability to learn to classify scene categories, and the properties of the learned representational space? Broadly, the term “Metameric” in Fig. 5A alludes to the \textit{concept} that under certain viewing conditions, the 3 type of image stimuli (given the nature of their distortions) could be metameric to a human observer given a wide field of view such as $40\deg\times40\deg$, a 100 ms blink and noise masking while holding center fixation. Recall we do not make any conclusions about metamers, but more about texture-like transformation in the periphery of a system. We have clarified this in our last revision (Section 1 + 2 including figures) and have answered this point raised by R4 as well!
> >
> > - In addition, “(2) I am not sure about the purpose of the "matched-resource" control and why SSIM is the right metric to use here. If it's about information content in the image, I think it should be evaluated in terms of mutual information between the image and the labels (which is hard); if it's about information available to human observers, then a perceptual metric is the right approach, but SSIM is a very poor one that doesn't correspond well with human perception. Therefore, I am skeptical that these baselines are really useful.”
> >
> > This is an interesting observation by R1, and you are right that exactly how you match these makes different equivalences. As previously noted, our aim wasn't to make subtle metamers for humans but to create accentuated foveated images for the models, so we could characterize the impact of this manipulation relative to non-foveated images and quantify these differences with a rate-distortion optimization for the Ada-Gauss-Net and Matched-Net controls. In this way SSIM (vs Mean Square Error or others) is a good choice because the perceptual metric is upper bounded, depends on contrast, luminance and local structure, and will take into account differences in resolution (important for Ada-Gauss-Net & Matched-Net). In addition, it has the added benefit that it is monotonically related to human perception and has been empirically verified with human judgements. Please refer to Supplementary Material figures 10 and 11 in our updated manuscript for a more detailed walk-through of this perceptual optimization pipeline where we use global and foveated derivations of SSIM. Furthermore, we did in fact want to render images at the “matched resource” stage but at the perceptual level, hence ‘perceptually resource matched’ (Foveation-Net, Matched-Net, Ada-Gauss-Net). I think here we both agree on the same thing, and have updated our submission to make this more clear.

---

> > > ### Author Response · Authors · 2020-11-19
> > > **Comments to AnonReviewer1 [3/4]**
> > >
> > > - Finally, “(3) The claim “greater iid generalization, high spatial frequency sensitivity and robustness to occlusion emerged exclusively in our foveated texture-based models” seems to be an overstatement. There appears to be basically no difference between foveated and standard net. In all Figures (5–8) the differences are mostly within the range of 1 SD and the order of the two models changes sometimes. The only effect that might be significant is the difference in the scotoma condition for intermediate to high occlusion conditions. The only robust difference is between standard+foveation and MatchedNet+AdaGauss, where the latter seem to carry much less information (see points (1) and (2) above).”
> > >
> > > [IMPORTANT] Here once again, perhaps we have misexpressed ourselves which has led to a confusion on R1’s side that we will take the time to dissect. Mainly: Foveation-Net can really *only* be compared to Matched-Net and Ada-Gauss-Net, NOT to Standard-Net. Standard-Net presents a perceptual upper bound of a system that has unfairly more available perceptual resources to perform inference (See Section 2.3 and Figure 6 - left). This is the main motivation of why we performed a perceptual optimization procedure to define a non-foveated system like Matched-Net and another foveated system that uses adaptive gaussian blurring (Ada-Gauss-Net). All the gold curves (Standard-Net), should be seen as perceptual upper bounds, and in a way show a great thing: even if you alter the image grossly in the periphery, you can do just as good as if you do not (navy curves close to gold as R1 confirms). The only situation where this breaks (for better!) is the robustness to occlusion experiment where we found that the actual crowding operator itself helped improve the robustness beyond the Standard-Nets because it would help fill in the occluded area. For all curves (as R1 also agrees) Foveation-Nets is clearly better than Matched-Net and Ada-Gauss-Net, this is indeed our goal, and what we want to show. Standard-Net is the `Supporting Actor’ that reminds us of what the upper bound of the other 3 perceptually matched resource systems (Foveation-Nets, Matched-Nets and Ada-Gauss-Nets) would be if more perceptual resources were accessible. R2, has articulated this quite well: “The paper explores how preprocessing an image with a foveated texture-based rendering - where content in the periphery is transformed in a lossy manner but still perceptually equivalent - affects the training of a downstream neural network. **It is shown that when equated to other lossy image transforms in terms of overall rate, that the foveated system shows better robustness to occlusion, generalization, and preservation of high spatial-frequency content.**”
> > >
> > > [IMPORTANT] Here is where maybe R1 (to play Devil’s advocate) may raise a perfectly valid concern: “Then why pre-process the image at all?”. A potential relevant  follow-up question that is worth clarifying because that is NOT what we are proposing (i.e. our goal is not to achieve SOTA on an accuracy benchmark). Rather, our paper studies the representational short-comings of spatially-uniform processing models and a “thorough empirical analysis” (as expressed by R4’s positive review) of foveated vs non-foveated models. More specifically, the questions we want to answer here are *if* a foveated transform alters the performance / robustness / biases / representations of a neural network: 1) by how much? (Foveation-Net vs Standard-Net); 2) how does re-distributing the resources change perceptual behaviours when the perceptual compression rates are equalized, and which one is the optimal one? (Foveation-Nets vs Ada-Gauss-Nets vs Matched-Nets); and 3) does the type of foveation transform matter at all (is texture better than blur)? (Foveation-Net vs Ada-Gauss-Net). We have emphasized this more with a new figure in our updated manuscript (Figure 5 (A)) in addition to bringing this up at Section 2.3 . Thank you R1!

---

> > > > ### Author Response · Authors · 2020-11-19
> > > > **Comments to AnonReviewer1 [4/4]**
> > > >
> > > > - “Sec 3.4: The connection between Gaussian blur and scale invariance is not clear”.
> > > >
> > > > Work by Poggio, Mutch & Isik (CBMM Memo 17, 2014) and Han, Roig, Geiger & Poggio  (Nature Scientific Reports, 2020) on eccentricity dependent neural networks (ENN’s) which use multi-resolution crops of a stimuli (analogous to multiple levels of a gaussian pyramid (Burt & Adelson, 1983; IEEE Transactions on Communications)) has explored a way to account for scale invariance during training in these deep networks, and that this eccentricity dependent mechanism is similar in humans that have varying size receptive fields in the periphery with gabor-like filters that vary in standard deviations of the gaussian component. Also see recently: Reddy, Banburski, Pant & Poggio (NeurIPS 2020) for how eye-movements/fixations can act as a proxy of multi-resolution sampling of an object when it is placed in the foveal and peripheral region, and how this can *increase* adversarial robustness.
> > > >
> > > > - Sec 3.4: The following statement is trivial, since the images were low-pass filtered: “We found that Foveation-Nets and Standard-Nets were more sensitive to High Pass Frequency information, while Ada-Gauss-Nets and Matched-Nets were sensitive to Low Pass Frequency stimuli”
> > > >
> > > > Not necessarily, it could have been that the crowding mechanism (that although still sharp appears distorted) in the periphery of the image biases the CNN to learn a larger number of low-pass frequency tuned filters (gabors with wider standard deviation). Recall that the CNN is trained with a weight-sharing constraint (both the AlexNet and ResNet18 backbones as shown in the Supplementary Material), so again, this is not necessarily trivial: it could have been the case that the weight-sharing constraint forced the collection of stage 2 deep networks to pick-up on the peripheral cues and that SGD biased the weights to be a family of low-pass filters -- but this was not the case. Instead, high pass-frequency filters were dominant for Foveation-Nets and low-pass frequency filters were dominant for Ada-Gauss-Net (*even when they both have the same high resolution fovea* -- recall the fovea of Ada-Gauss-Net is matched to Foveation-Net and is *not* blurred). Training foveated networks that have a crowding-operator in the visual periphery without weight sharing constraints similar to Yu & Konkle, Vision Sciences Society 2017 (and Wang & Cottrell, Journal of Vision 2017) is an interesting direction of future work. For Matched-Nets however R1 is correct, the results would be trivial, what is surprising however is how similar Matched-Nets and Ada-Gauss-Nets behave even when Ada-Gauss-Nets does preserve its high spatial frequency fovea.
> > > >
> > > > - Fig 8: It is unclear how it argues for shape bias.
> > > >
> > > > Please see our reply linked here to Reviewer 3 who asked the same question:
> > > > Link: https://openreview.net/forum?id=2_Z6MECjPEa&noteId=XZZqcZkRXsk
> > > >
> > > > - Definition of "foveation" seems strange
> > > >
> > > > We’d like to re-clarify our use of the term foveation, which draws on vision science concepts to take a high-resolution image (akin to the world), and transform it into a spatially-varying representation which has high-resolution foveal information, and increasingly distorted/texturized information in the periphery,  inspired by both the anatomy of the retina and cortex, as well as the hypothesized texture-representation and crowding mechanisms of the visual periphery. Note: foveation and fixation are not the same thing, and they have been interpreted as misnomers in computer vision & ML (and in some cases even confused with attention; (See Lindsay 2020, Frontiers in Computational Neuroscience). “To fixate” and “to foveate” can indeed be used interchangeably to convey maintaining an observer's gaze on a specific location (Eckstein, 2011; Journal of Vision), however “foveation” is classically defined as spatially-varying visual processing from works in image processing dating as far back as Perry & Geisler, SPIE Proceedings, 1998: “A real-time foveated multiresolution system for low-bandwidth video communication”, to recently in computer graphics: Kaplanyan et al.’s: “DeepFovea: neural reconstruction for foveated rendering and video compression using learned statistics of natural videos”, ACM Transactions on Graphics, 2019.
> > > >
> > > > - "Matched-resource" controls are somewhat questionable
> > > >
> > > > Addressed earlier in Reply Part [2/4].
> > > >
> > > > - “No evidence provided that "foveated" images are actually metameric”
> > > >
> > > > Addressed earlier in Reply Part [2/4].
> > > >
> > > > - “Unclear if the premise of the paper is well-conceived”
> > > >
> > > > Addressed throughout the reply.

---

> > > > ### Comment · AnonReviewer1 · 2020-11-21
> > > > **Not convinced**
> > > >
> > > > Thank you for the explanation. However, I am not convinced by these arguments, as they seem to partly contradict your first response. If the goal is not to compete with the unprocessed image, why do you bring up stylized ImageNet et al., all approaches meant to improve robustness for real-world scenarios.
> > > >
> > > > Leaving that aside, even if we treat the standard net just as an upper bound, I’m not convinced the two control models are terribly useful for the reasons I explained. Your foveated net gets a full image as input and it’s not clear from an information-theoretic point of view how much information has been removed compared to the original. SSIM as a perceptual metric (and not a particularly good one) just doesn’t seem to be a very useful metric here.

---

> > > > > ### Author Response · Authors · 2020-11-25
> > > > > **Resolving follow-up comments**
> > > > >
> > > > > - "Thank you for the explanation. However, I am not convinced by these arguments, as they seem to partly contradict your first response. If the goal is not to compete with the unprocessed image, why do you bring up stylized ImageNet et al., all approaches meant to improve robustness for real-world scenarios."
> > > > >
> > > > > We think that foveation has the potential to improve robustness, akin to the stylized Imagenet -- though currently our data only show slightly better o.o.d. generalization and we are currently investigating adversarial robustness. This paper lays important groundwork for thinking about systems that do have spatially-varying resolution/computation cameras *(with a resource constraint, as happens in the real world).*  For example, having a foveated visual system may provide long-term positive impact on generalization and robustness and other perceptual biases for drone navigation, autonomous driving, or scene recognition on phone cameras (we empirically show Foveation-Net > Matched-Net).
> > > > >
> > > > > - "Leaving that aside, even if we treat the standard net just as an upper bound, I’m not convinced the two control models are terribly useful for the reasons I explained. Your foveated net gets a full image as input and it’s not clear from an information-theoretic point of view how much information has been removed compared to the original. SSIM as a perceptual metric (and not a particularly good one) just doesn’t seem to be a very useful metric here."
> > > > >
> > > > > This is a great point, and we initially had a similar stance as yours in earlier versions of this work, we thought it may have not been necessary, but in order to properly verify the contributions of a texturized periphery we needed to weed out the contributions that texture was buying us, and if we could have arrived to the same conclusions with only adaptive gaussian blurring (hence the Ada-Gauss-Net control). Suppose we found that Ada-Gauss-Nets delivered very similar curves as Foveation-Nets, then our conclusions would have changed dramatically, and we could not have been able to dissociate the contributions of texture vs blur in the periphery. Indeed, in our paper we found: texture giving greater i.i.d generalization, while adaptive blur greater o.o.d generalization; texture giving significantly greater robustness to occlusion and high spatial frequency bias than adaptive blur; however it seems that independent of texture or blur, either foveated system learns a center image bias; recall *Foveation-Nets and Ada-Gauss-Nets have the same high acuity fovea*. As for Matched-Net the reasoning lies in understanding what would happen if we had a system with the same amount of perceptual resources as Foveation-Net that have been 'lost' in the texturization process, but that is non-foveated. This control would then help to see if any contributions of Ada-Gauss-Net (such as greater o.o.d generalization) are purely driven due to the blur itself or the fact that the blur was adaptive (hence foveated).
> > > > >
> > > > > Ultimately, you are correct that it is non-trivial to make information-theoretical guarantees with respect to perceptually matching these systems (Foveation-Net, Ada-Gauss-Net, Matched-Net). We propose one solution via the Rate-Distortion optimization of Balle et al. 2017 and the SSIM-driven optimization of Deza et al. 2019 (which has found psychophysical promise for this metric for these types of experiments), but you are correct that the values/strengths of Gaussian blur for either Ada-Gauss-Net or Matched-Nets would change depending on the choice of metric and optimization. There are hundreds of perceptual metrics however, and choosing one over the other is not obvious (we chose SSIM for reasons mentioned in our rebuttal), in any case we think this is a first step towards a practical analysis of perceptual bounds of foveated systems and comparing foveated systems to each other. In the future, we agree with you that having an information-theoretical framework for an optimal perceptual resource matching procedure would be a great step forward (and it is a great research question in itself!); we will add this in our final version of the paper following all the exchange in the discussions.

---

> > > ### Comment · AnonReviewer1 · 2020-11-21
> > > **What’s the purpose of the matched resource controls?**
> > >
> > > Thank you for the response and clarifying that the images are not actually metameric.
> > >
> > > Regarding the matched resource controls, I’m still in the dark. Could you state again clearly what you need it for? It just doesn’t become clear to me what MatchedNet and AdaGaussNet contribute. Which conclusion could you not draw if the two controls were not included in the paper?

---

> > > > ### Author Response · Authors · 2020-11-25
> > > > **Resolving follow-up comments**
> > > >
> > > > - "Thank you for the response and clarifying that the images are not actually metameric."
> > > >
> > > > Thanks for pointing out the confusion! The paper is much clearer for it.
> > > >
> > > > - "Regarding the matched resource controls, I’m still in the dark. Could you state again clearly what you need it for? It just doesn’t become clear to me what MatchedNet and AdaGaussNet contribute. Which conclusion could you not draw if the two controls were not included in the paper?"
> > > >
> > > > See response to the comment that begins with “training with stylized networks…” here: https://openreview.net/forum?id=2_Z6MECjPEa&noteId=gywibux93s7 And to “Leaving that aside, even if we treat the standard net just as an upper bound, I’m not convinced the two control models are terribly useful for the reasons I explained.”, for a more thorough discussion here: https://openreview.net/forum?id=2_Z6MECjPEa&noteId=eWEdiCtUmCR

---

> > ### Comment · AnonReviewer1 · 2020-11-21
> > **Not convinced**
> >
> > Thank you for your response. Unfortunately I can’t follow. You say foveation provides a “functional benefit.” Could you point to the figure that shows foveation to be beneficial compared to just processing the plain image (i.e. standard net)?
> >
> > Training on stylized images, to pick one of your examples, is different: in this case the network is trained on stylized images, but no stylization is used during inference. Thus, the net learns to rely on different features during training and the authors show that nets trained on such way generalize better to image perturbations than CNNs trained on plain ImageNet. In your case I don’t see such a demonstration. As I stated in my original review, in all figures it looks like performance of standard net and foveation are essentially the same. What am I missing?

---

> > > ### Author Response · Authors · 2020-11-25
> > > **Resolving follow-up comments**
> > >
> > > - "Thank you for your response. Unfortunately I can’t follow. You say foveation provides a “functional benefit.” Could you point to the figure that shows foveation to be beneficial compared to just processing the plain image (i.e. standard net)?"
> > >
> > > Hi R1, thanks for your follow-up comments! Before we precisely address the comparisons of Foveation-Net vs Standard-Net, we’d like to clarify that we do not make a claim of Foveation-Net *having greater accuracy or i.i.d generalization* than Standard-Net -- which is consistent with your information-theoretical arguments. Indeed, Foveation-Net comparisons should be made in reference to the other perceptually matched systems as stated before: Matched-Net and Ada-Gauss-Net, as Standard-Net has more perceptual resources. We have fixed the language in our updated manuscript that may suggest an accuracy-related benefit of Foveation-Net.
> > >
> > > As for your question, the figures that *do* show a functional benefit (with respect to Standard-Net) is Figure 7A top left (for scotomal occlusion given the nature of the foveation transform that implicitly manipulates occluded area due to crowding and in consequence boosts performance), and also Figure 6B when performing o.o.d generalization exclusively between Foveation-Nets vs Standard-Nets. Importantly, there was little ‘cost’ in terms of scene categorization accuracy for Foveation-Net (which is interesting, but not an ‘accuracy benefit’ per se -- that again we do not make a strong claim about, and we both agree on). Instead the consequences of the foveation module seem to be in the nature of the learned representation, e.g. showing more sensitivity to information in the center of the image, and relatively more sensitivity to high than low spatial frequency, compared to Standard-Nets; in some cases this *could* be a functional benefit that does not need to be reflected through accuracy.  Whether these representational signatures confer a functional benefit for other tasks is open (such as to adversarial robustness), and we have revised the current manuscript to reflect this logic.
> > >
> > > Finally, we’d like to clarify that Standard-Net is *not* the network trained on the un-processed / plain image. Standard-Net is the network trained on the images sent through the Foveation Transform (a convolutional auto-encoder) with scaling factor (and thus the distortions) set to 0 (See section, 2.3) -- hence the need for a perceptually based rate-distortion optimization procedure for the Matched-Net and Ada-Gauss-Net controls. This form of a control yields a *tighter* upper bound than training/testing on unprocessed images.
> > >
> > > - "Training on stylized images, to pick one of your examples, is different: in this case the network is trained on stylized images, but no stylization is used during inference. Thus, the net learns to rely on different features during training and the authors show that nets trained on such way generalize better to image perturbations than CNNs trained on plain ImageNet. In your case I don’t see such a demonstration. As I stated in my original review, in all figures it looks like performance of standard net and foveation are essentially the same. What am I missing?"
> > >
> > > You are reading the graphs correctly, performance (accuracy) of Foveation-Net is essentially the same with Standard-Net, with different representational signatures.  Our other models, work to triangulate the mechanisms that are accounting for these differences: 1- is it simply the spatially varying pooling windows? 2- would the same principles emerge from a network that is also foveated ‘discards information’ from the input in a different way than Foveation-Net, but to a similar degree (as estimated with a rate-distortion optimization).  These other models do show greater costs relative to Standard-Nets -- in this way we can see that the foveation transform is having a consequence on the representation, effectively preserving classification ability but shifting the information that is being relied on to do so.
> > >
> > > We elaborate more on this in your other follow-up comments here: https://openreview.net/forum?id=2_Z6MECjPEa&noteId=eWEdiCtUmCR

---

### Official Review · AnonReviewer2 · 2020-10-31
**A nice demonstration of the advantages of foveated texture-based image preprocessing**

**Rating:** 7
**Confidence:** 3

**Review:**

The paper explores how preprocessing an image with a foveated texture-based rendering - where content in the periphery is transformed in a lossy manner but still perceptually equivalent - affects the training of a downstream neural network.  It is shown that when equated to other lossy image transforms in terms of overall rate, that the foveated system shows better robustness to occlusion, generalization, and preservation of high spatial-frequency content.

The many figures are difficult to wade through because the are so compressed and cluttered with many labels and diagrams.  I would recommend figuring out a way to prune these down so they are more readable and convey just central punchline that you are trying to show.

---

> ### Author Response · Authors · 2020-11-19
> **Comments to AnonReviewer2 [1/1]**
>
> - The paper explores how preprocessing an image with a foveated texture-based rendering - where content in the periphery is transformed in a lossy manner but still perceptually equivalent - affects the training of a downstream neural network. It is shown that when equated to other lossy image transforms in terms of overall rate, that the foveated system shows better robustness to occlusion, generalization, and preservation of high spatial-frequency content.
> - The many figures are difficult to wade through because they are so compressed and cluttered with many labels and diagrams. I would recommend figuring out a way to prune these down so they are more readable and convey just central punchline that you are trying to show.
>
> Thank you for this feedback R2! We have just added an additional key figure with a summarized Methods section on page 5. The goal of this figure and summarized Methods section is to give the reader a breath and present a high-level overview (and legend -- for the figures) that should implicitly make all the other plots more accessible and intuitive. Follow-up feedback on this edit is much appreciated.

---

### Official Review · AnonReviewer4 · 2020-11-02
**clearly defined model; thorough experimental analysis; clarity could be improved**

**Rating:** 7
**Confidence:** 4

**Review:**

##########################################################################
Summary:

In this paper, authors study the functional advantages of a foveal transform of visual inputs. It is nicely introduced with a very comprehensive review of the literature. The method introduces a 2 two stage model of the visual system, where the first stage corresponds to the (fixed and non adaptive) foveation stage and the second stage to the higher level processing, typically associated with the categorisation operated in the ventral stream of the visual pathway. This second stage will be implemented by existing CNN architectures (AlexNet and ResNet) which are re-learned on the transformed inputs. To control for the functional consequences of the foveated processing, the first stage can also be a single isotropic blurring of the image. Both alternatives are manipulated such that their distortion (as computed with a SSIM measure) are equally balanced, leading to 1 standard mapping and three proposal retinal transformations :  Standard-NEt (unmatched) and Foveation-Net, Matched-Net and Ada-Gauss-Net (matched). Results show that for both perceptual systems which are foveated, "Foveation-Net has the highest i.i.d generalization while Ada-Gauss- Net has the greatest o.o.d generalization". Second result is that foveated processing allowed a better robustness to occlusions and third result is that such networks reproduce behavioural results of a Window Cue-Conflict. Last results propose to study that foveation introduces a focusing strategy, and keep high-french information on the fovea - which seem less striking results.

Overall, the paper is original, technically sound and very well supported by experiments. However, there are some concerns that I highlight below.

##########################################################################
Concern:

The  paper is very dense and wants to say too much while perhaps losing on the main point: the foveal transform In particular, there is one point about foveation and another about metamerism (and crowding). While the first is studied in depth, the second is not studied fully, or at least not parametrically. This hinders the comprehension of the mechanisms that may be at play behind the functions of foveation.  Controlling for the amount of "metameric distortion" at different eccentricities both in models and humans would be a novel contribution to the field.

More generally, the paper seems to explore many computational alternatives but perhaps misses a key property of the visual system, that is that RFs size grow in size AND density as a function of eccentricity - it seems the metameric distortions adds a novel ingredient as it tunes the precision of textural information as a function of eccentricity. taken together, this factors could simplify the presentation of the paper by presenting different hypothesis for the evolution of RF size and density and in particular to show why a given magnification factor may be better than other. I consider the iid vs ood and occlusion results to be the most convincing, but at the end of reading the paper, I lack a comprehension of why we have such results, and what hyperparameters play the most important role.

More particularly, for the robustness to occlusions, the results are in my understanding given of the systems learned in the non-occluded case. Would you predict that results would generalise if you re-learn weights from the occluded images? The Window Cue-Conflict results are less convincing  as the effects are rather small. Perhaps using different hyper parameters (magnification ratio, metameric distortion) would help? The conclusion "that foveation (in general) seems to induce a focusing mechanism (...) while the texture-based computation still preserves high spatial-frequency selectivity" seem to be a mechanical consequences of the foveation transform (you focus on a point while preserving cone density in the fovea) - did I miss something?  Finally, I know the problem of page budget, but I had to go page 16 of the SM to have the definition of the dataset and get eg the value of the chance level - a synthetic overview on p.5 would be welcome.

Concerning the form of the paper, clarity could be improved in some sections. Some sentences (e.g. "same pattern of results are evident for ResNet18" (p. 5) "an unexpected victory for Foveation-Nets" (p.5)  "is still quite impressive" (p.6), ...) are more reminiscent to stunts from the PR department than is necessary in a scientific paper. Please avoid.
Last, missing the code
##########################################################################

minor
p.4 « guassian » _

---

> ### Author Response · Authors · 2020-11-19
> **Comments to AnonReviewer4 [1/3]**
>
> Thank you for your positive feedback Reviewer 4! We will address your additional concerns in the following replies.
>
> - “The paper is very dense and wants to say too much while perhaps losing on the main point: the foveal transform In particular, there is one point about foveation and another about metamerism (and crowding). While the first is studied in depth, the second is not studied fully, or at least not parametrically. This hinders the comprehension of the mechanisms that may be at play behind the functions of foveation. Controlling for the amount of "metameric distortion" at different eccentricities both in models and humans would be a novel contribution to the field.”
>
> Thank you for raising this point about clarity -- in fact you are right that while foveation is what we focused on, we in fact didn’t study metamerism (e.g. perceptual invariance) directly.  Indeed, our intention was not to render metamers for humans per se, but to use the spatially-varying distortions (as used in the metamers of Freeman & Simoncelli, 2011), to overemphasize and explore the impact of a texturized content in the periphery on the properties of the subsequently learned representation. We also wanted to make clear that when thinking about foveation, one can think outside of the box beyond adaptive gaussian blurring. We have clarified this point in the revised paper (see Section 2.1; updated Figure 2 + 6 that used ‘Metameric”, and added a new Figure in page 5) and will continue to refer to the `Metamer Model’ as a foveation transform as in our initial submission.
>
> - “More generally, the paper seems to explore many computational alternatives but perhaps misses a key property of the visual system, that is that RFs size grow in size AND density as a function of eccentricity - it seems the metameric distortions adds a novel ingredient as it tunes the precision of textural information as a function of eccentricity. taken together, this factors could simplify the presentation of the paper by presenting different hypothesis for the evolution of RF size and density and in particular to show why a given magnification factor may be better than other. I consider the iid vs ood and occlusion results to be the most convincing, but at the end of reading the paper, I lack a comprehension of why we have such results, and what hyperparameters play the most important role.”
>
> This is a great point! We have thought a lot about potentially re-running these experiments on different scaling factors of the metamer model ranging including values of 0.3 and 0.5, or even as aggressive as 0.7 (0.4 was used in our paper; Specifications discussed in Supplementary Material Section 6.1 of updated manuscript). This is a direction of on-going work that we did not want to put in this paper as it would have required us to also run more controls (Matched-Nets and Ada-Gauss-Nets for every scaling factor used). Once again, it is an exciting idea that we considered a priori, yet the goal of this paper is to be a chapter of a series of questions pertaining the role of foveation (specially of the texture-like type) in humans and machines. As a side-note, you may also want to see in the Supplementary Material where we explored the effects of eye-movements for Foveation-Nets and data-augmentation for Standard-Nets. These results were included in the Supplementary Material of our original submission. We decided to put these results in the Supplement (and refrain from exploring your great suggestion above) as we did not want to clutter the paper with too many ideas as you pointed out earlier in your review.

---

> > ### Author Response · Authors · 2020-11-19
> > **Comments to AnonReviewer4 [2/3]**
> >
> > - “More particularly, for the robustness to occlusions, the results are, in my understanding, given of the systems learned in the non-occluded case. Would you predict that results would generalise if you re-learn weights from the occluded images?
> >
> > We are not sure how each system would perform when trained on occluded images, and this is an interesting follow-up experiment to run for future work given that it would bias each perceptual system to learn the cues either in the non-occluded region (be it the periphery, or the fovea depending on the occlusion). We suspect that we would see a very similar pattern of results with Foveation-Nets on top of Matched-Nets and Ada-Gauss-Nets, if the task is *scene recognition* given that in some cases scene recognition can be driven by texture priors (Renninger & Malik, 2004). In the realm of object recognition works such as CutOut (DeVries & Taylor. ArXiv 2017) has shown that training a CNN on occluded CIFAR-10 images of random patch size and *on random locations* can indeed improve performance as it acts as a regularizer, it would be interesting to see if the same procedure (coupled with the Foveation Transform) would give similar results. We suspect performance across all models would systematically increase mildly but it’s quite likely that the center image bias for foveated trained networks would go away. See also Tsank & Eckstein (The Journal of Neuroscience, 2017) “Domain Specificity of Oculomotor Learning after Changes in Sensory Processing” for experiments like these which were done on humans and computational models (foveated ideal observers) that eventually find ‘new’ optimal points of fixation. Once again, many follow-up questions arise and we are excited to see these suggestions as avenues of future work for the community!
> >
> > - “The Window Cue-Conflict results are less convincing as the effects are rather small. Perhaps using different hyper parameters (magnification ratio, metameric distortion) would help? The conclusion "that foveation (in general) seems to induce a focusing mechanism (...) while the texture-based computation still preserves high spatial-frequency selectivity" seem to be a mechanical consequences of the foveation transform (you focus on a point while preserving cone density in the fovea) - did I miss something?”
> >
> > You are right, the effects are small (as they are area ratios of approx 0.37 to 0.43 ) but are statistically significant when performing comparisons for area under the curve for the peripheral or foveal bias -- from which the cross-over points are derived. Further, recall that this center image bias is non-trivial: it could have been the case that the texturized representation of the periphery was a better signal for scene content, and that these scene-classifying foveated networks would leverage the peripheral content more than the foveal one. However, this wasn't the case. Instead the foveation transform seemed to provide a natural inductive bias towards the central region of the image. In this case, it is a mechanical consequence in some way, but the functional consequence of this foveation stage on scene representation was not known.
> >
> > Also, notice that Foveation-Net and Ada-Gauss-Net (both foveated with different type of processing mechanisms: texture or blur) have very similar cross-over points compared to Standard-Net and Matched-Net (which are both non-foveated). So when we argue “foveation (in general), seems to induce a focusing mechanism” we mean that any of the 2 types of spatially varying computation (texture or adaptive blur) will induce a center localized bias.
> >
> > [IMPORTANT] With regards to high-spatial frequency selectivity, Figure 9 B in the updated manuscript shows the peculiarity that *even when* Foveation-Net and Ada-Gauss-Net *both* have the same preserved high spatial frequency fovea, *only* Foveation-Net preserves high-spatial frequency selectivity, likely due to weight sharing constraints in learning for the CNN in stage 2. This is a non-trivial consequence and please see our extended discussion replying to R1’s similar concern as well, link: https://openreview.net/forum?id=2_Z6MECjPEa&noteId=W2Kw2dL3Uzf

---

> > > ### Author Response · Authors · 2020-11-19
> > > **Comments to AnonReviewer4 [3/3]**
> > >
> > > - “Finally, I know the problem of page budget, but I had to go page 16 of the SM to have the definition of the dataset and get eg the value of the chance level - a synthetic overview on p.5 would be welcome.”
> > >
> > > Agreed, we used our additional main page in the body to have a summarized Methods section and introduced the dataset (page 5). Please feel free to provide any follow-up feedback with regards to the extra figure and page.
> > >
> > > - Concerning the form of the paper, clarity could be improved in some sections. Some sentences (e.g. "same pattern of results are evident for ResNet18" (p. 5) "an unexpected victory for Foveation-Nets" (p.5) "is still quite impressive" (p.6), ...) are more reminiscent to stunts from the PR department than is necessary in a scientific paper. Please avoid. Last, missing the code
> > >
> > > Thanks R4, we will improve on clarity and rephrase such claims, and please feel free to check the updated manuscript that we have uploaded!

---

### Official Review · AnonReviewer5 · 2020-12-04
**Interesting study to assess the functional benefit of foveated perception**

**Rating:** 5
**Confidence:** 4

**Review:**

I thank the authors for a thoroughly written paper studying an important question for both machine learning and neuroscience. The authors propose a biologically inspired modification to CNN architectures by introducing foveation. Several thorough experiments are performed to assess the benefit of foveation with reasonable control transformations. The proposed modifications seem novel and discuss the relevant prior work in this domain. Appreciate the detailed discussion of all implementational specifics, I'm fairly confident about the correctness of the experiments performed and results presented.

# Some concerns I have about the claims:
I agree with some of R1's comments, the claims seem to be overstated in my opinion. In Fig. 7, it looks like Foveation-net significantly outperforms other baselines only in pre-foveation scotoma transform. This is likely because compared to Standard-Net, Foveation-Net has larger unoccluded information due to the nature of the foveation transform. In the post-foveation scotoma condition wherein the amount of unoccluded area is matched, Standard-Nets outperform Foveation-Net. This is fine, but I didn't read any discussion on this observation in the submission. I see that the authors have attempted to address this question in other comments below. In summary, I doubt whether foveation truly provides more robustness to occlusions in the presented sense.

Fig. 9C does not seem like sufficient evidence to claim that foveation promotes shape bias over texture bias.

As mentioned earlier, find the topic studied to be a very interesting and an important one. I'm certain this work will stimulate interesting discussions and future work to better understand the functional significance of foveation. The analyses are thorough and the reported results seem to be accurate. However I doubt whether the proposed claims are justified by the analyses. The figures are great, particularly the ones that describe procedures like foveation, scotoma, etc. The ones analyzing model performance seemed a little shrunk and I had to zoom in to glean the details. I would suggest moving some/all figures that discuss the foveation/occlusion procedures (such as in Figs 7&8) to SI to give more room for the quantitative analysis figures.

# Questions to authors:
1. What are your thoughts on applying foveation-like masking throughout the network's activations and not just at the input stage?
2. I'm very interested to know whether the proposed mechanism can buy more performance on Stylized ImageNet, and robustness to adversarial perturbations.
3. The authors might be interested to explore how well foveated model performs on datasets like CIFAR-10C and ImageNet-C.

---

### Author Response · Authors · 2020-11-19
**Comments to All Reviewers**

Thanks to all reviewers for your time, comments and feedback! It seems like most reviewers have had a very positive initial assessment of our paper (7,7,7), with the exception of Reviewer 1 (3), who raises very interesting points that we will discuss in detail below and also encourage the other reviewers to read as it may potentially answer any lingering questions/thoughts. In general, we are delighted to see that our idea of exploring the texture-based implications of a crowding-based foveation transform is well received and that it can potentially create a positive impact for both machine + human vision through ICLR. Our code which includes all 120 (80 in the main body of paper + 40 in the Supplementary Material) trained AlexNet and ResNet18 models in PyTorch, the original & modified (foveated) training and testing image datasets, and the data analysis, will be available online if the paper is accepted to replicate and build upon these results for the larger Computer Vision, Machine Learning and Neuroscience community. We will add all additional citations included in this on-going Discussion in our final version of the paper.

In addition, we have updated our manuscript where we addressed several typos and relaxed the tone of our claims, but mainly introduced *page 5* which gives a breather and provides a general/summarized overview of the Methods section and also introduces 2 key sub-figures that make the flow of the rest of the paper more digestible. The first, is a color-coded sub-figure that gives insight to the role of each perceptual system and how (when paired together) answer different questions, and the second  is a sub-figure of the used scene dataset.

---

### Decision · Program_Chairs · 2021-01-07
**Final Decision**

**Decision:**

Reject

**Comment:**

This paper explores the use of a texture-based foveation stage in
scene categorization.  They show that the foveated system shows
presevation of high spatial-frequency information relative to other
matched transformations.

This paper engendered a lot of discussion and had a wide range of
ratings.  An extra review was requested that fell intermediate between
the high and low scores.  Generally reviewers agree that the question
is interesting but that the paper does not clearly elucidate the logic
of the paper.  That is, Reviewer 1, and another reviewer in
discussion, had issues with the logic of the paper. I also found the
motivation behind the paper, difficult to understand at first.  I had
to find the Rosenholtz paper to understand why the authors were
considering this particular representation. This should be better
explained in the paper - the main points of the paper should be
understandable by itself.


There is also concern that the claims are not validated by
the presented results.  For example, the authors claim that their
foveated network were more robust to occlusion but Reviewer 5 points
out that this is likely due to the foveation-nets having more
unoccluded information.


On the positive side, Reviewer 4 points out that the experiments are
extensive and several reviewers commented that they trust that the experiments
were done correctly.  Reviewer 2, 4 and 5 all mention that there are too many
results reported and recommend paring down to the most important
results.

In my view the paper is right at the border of acceptance.  Acceptance/rejection will
depend on capacity limits and balancing areas.  I recommend that if accepted,
or resubmitted to another conference, that the results be pared down, and
more space be devoted to explaining the question(s) and why they are
interesting and relevant and why the comparison networks allow the questions
to be answered.

Originality - High
Quality - High
Clarity - Could be improved
Significance - Could be better articulated and is hard to assess as is.
Pros: - interesting idea, many well done experiments
Cons: - claims not well validated, clarity could be improved to	emphasize significance
Other: paper too dense - should be pared down to improve clarity